# Macrophages take up VLDL-sized emulsion particles through caveolae-mediated endocytosis and excrete part of the internalized triglycerides as fatty acids

Lei Deng[1], Frank Vrieling[1], Rinke Stienstra[1,2], Guido J. Hooiveld[1], Anouk L. Feitsma[3], Sander Kersten[1]*

1 Nutrition, Metabolism and Genomics Group, Division of Human Nutrition and Health, Wageningen University, Wageningen, the Netherlands, 2 Department of Internal Medicine, RadboudUMC, Nijmegen, the Netherlands, 3 FrieslandCampina, Amersfoort, the Netherlands

* sander.kersten@wur.nl

**Data Availability Statement:** All sequencing data have been submitted to the Gene Expression Omnibus (GEO), and are available under accession

## Abstract

Triglycerides are carried in the bloodstream as part of very low-density lipoproteins (VLDLs) and chylomicrons, which represent the triglyceride-rich lipoproteins. Triglyceride-rich lipoproteins and their remnants contribute to atherosclerosis, possibly by carrying remnant cholesterol and/or by exerting a proinflammatory effect on macrophages. Nevertheless, little is known about how macrophages process triglyceride-rich lipoproteins. Here, using VLDL-sized triglyceride-rich emulsion particles, we aimed to study the mechanism by which VLDL triglycerides are taken up, processed, and stored in macrophages. Our results show that macrophage uptake of VLDL-sized emulsion particles is dependent on lipoprotein lipase (LPL) and requires the lipoprotein-binding C-terminal domain but not the catalytic N-terminal domain of LPL. Subsequent internalization of VLDL-sized emulsion particles by macrophages is carried out by caveolae-mediated endocytosis, followed by triglyceride hydrolysis catalyzed by lysosomal acid lipase. It is shown that STARD3 is required for the transfer of lysosomal fatty acids to the ER for subsequent storage as triglycerides, while NPC1 likely is involved in promoting the extracellular efflux of fatty acids from lysosomes. Our data provide novel insights into how macrophages process VLDL triglycerides and suggest that macrophages have the remarkable capacity to excrete part of the internalized triglycerides as fatty acids.

## Introduction

Lipids are essential for all cells, either as structural components, signaling molecules, or fuel source. Lipids are transported through the bloodstream as part of lipoproteins. Whereas cholesterol is mainly carried in high-density and low-density lipoproteins, triglycerides (TG) are predominantly transported by chylomicrons and very low-density lipoproteins (VLDL). Chylomicrons have a diameter of 75 to 600 nm, carry dietary TG, and are produced by enterocytes [1]. Conversely, VLDL have an average diameter of 30 to 80 nm, carry TG that are produced

number GSE203250. The FACS data is available in FlowRepository under the ID number FR-FCM-Z5KZ, FR-FCM-Z5K3, FR-FCM-Z5KY, and FR-FCM-Z5JV.

**Funding:** The work was supported by the PACMAN project (181102) funded by FrieslandCampina (https://www.frieslandcampina.com/) and a CSC Scholarship (201703250072) funded by China Scholarship Council (https://www.csc.edu.cn/). S. K. is the recipient of the PACMAN grant. L.D. is the recipient of the CSC grant. The funders had no role in study design, data collection and analysis, decision to publish, or preparation of the manuscript.

**Competing interests:** The authors have declared that no competing interests exist.

**Abbreviations:** AU, airy unit; ER, endoplasmic reticulum; GEO, Gene Expression Omnibus; LAL, lysosomal acid lipase; LPL, lipoprotein lipase; NPC2, Niemann–Pick type C2; RT-PCR, real-time polymerase chain reaction; siRNA, small interfering RNA; SLR, signal log ratio; TG, triglyceride; TRL, TG-rich lipoprotein; VLDL, very low-density lipoprotein; WLL, white light laser.

endogenously, and are synthesized in the liver [2]. In the fasted state, TG are present in the blood almost entirely as part of VLDL and its remnant lipoproteins.

Macrophages are innate immune cells that form the frontline in the host's defense against pathogens. They are specialized in the detection, phagocytosis, and destruction of bacteria and other harmful organisms. In addition, macrophages can present antigens and regulate inflammation by releasing cytokines. Besides phagocytizing and neutralizing pathogens, macrophages can also scavenge lipids, which after uptake can be stored in specialized organelles called lipid droplets. How macrophages scavenge VLDL and chylomicrons, and how the associated lipids are internalized and processed has not been well characterized.

Lipid uptake and storage in macrophages have primarily been investigated in the context of atherosclerosis [3]. Macrophages take up oxidized LDL, which is considered a key event in the pathogenesis of atherosclerotic lesions [4]. In the past decades, evidence has been accumulating that apart from LDL particles, TG-rich lipoproteins (TRL) and their remnants also contribute to atherosclerosis [5], possibly by carrying remnant cholesterol and/or by exerting a proinflammatory effect on macrophages [6]. Consistent with their purported roles in atherosclerosis, VLDL and VLDL remnants can be taken up and retained in the intima, where they can interact with macrophages [7].

The uptake of oxidized LDL and cholesterol by macrophages is mediated by a group of structurally unrelated molecular pattern recognition receptors referred to as scavenger receptors, including SCARB1, CD36, and MSR1 [8]. However, in contrast to the uptake of oxidized LDL, little is known about how macrophages take up and process VLDL particles. It has been shown that the uptake of VLDL-TG in cultured macrophages is promoted by lipoprotein lipase (LPL) [9]. Besides via its lipolytic function, LPL may enhance lipid uptake by functioning as a molecular bridge between VLDL and lipoprotein receptors and/or heparan sulfate proteoglycans [10,11]. Other steps in the uptake of VLDL-TG by macrophages remain poorly defined.

Endocytosis describes the transport of extracellular substances or particles into cells and is tightly related to the biological function of macrophages. The endocytosis pathway can be classified into several types: clathrin-dependent endocytosis, clathrin-independent endocytosis, pinocytosis, and phagocytosis [12]. The importance of clathrin-mediated endocytosis in the cellular uptake of lipids is well established. Indeed, the cellular uptake of LDL is mediated by the binding of LDL to the LDL receptor, followed by the formation of clathrin-coated pits and subsequent delivery of the lipid cargo to the lysosomes [13]. Receptor-mediated endocytosis also mediates the uptake of native or modified LDL and remnant lipoproteins in macrophages as a key step in the pathogenesis of atherosclerosis [14]. In hepatocytes, receptor-mediated endocytosis is required for the uptake of VLDL and chylomicron remnants [15]. Whether endocytosis contributes to the uptake of VLDL-TG by macrophages is unclear.

After uptake by cells, LDL is degraded in lysosomes, releasing free cholesterol. The cholesterol binds to Niemann–Pick type C2 (NPC2) before being shuttled out of the lysosome via NPC1 [16–18]. Part of the cholesterol transported via this pathway may go to the plasma membrane, while another portion may go to the endoplasmic reticulum (ER). The lysosomal membrane protein STARD3 transports the cholesterol directly from the lysosome to the ER by interacting with vesicle-associated membrane protein-associated protein (VAP)A/B on the ER membrane [19]. Interestingly, STARD3 also promotes the reverse transport of cholesterol from the ER to the lysosome [14], as well as the transfer of cholesterol from the lysosome to mitochondria [20]. While there is thus substantial insight into how cholesterol is processed, transported, and stored in macrophages, our mechanistic understanding of the processing of VLDL-TG in macrophages is very limited.

Considering the growing recognition of the importance of TRL and their remnants in atherosclerosis [5], here, using VLDL-sized TG-rich emulsion particles, we aimed to study the mechanism by which VLDL-TG are taken up and processed in macrophages.

## Results

### VLDL-sized lipid emulsion particles are taken up by macrophages

Using a microfluidizer, TG-rich emulsion particles were prepared with a mean diameter of 60 nm, referred to as VLDL-sized emulsion particles (Fig 1A) [2]. Treatment of human primary macrophages with these particles for 6 hours led to marked lipid accumulation, as visualized using BODIPY 493/503 staining (Fig 1B) and quantified using flow cytometry (Fig 1C). Treatment with VLDL-sized emulsion particles also significantly increased the expression of lipid-sensitive genes as measured by real-time qPCR (Fig 1D). Heatmaps (Fig 1E) and volcano plots (Fig 1F) based on RNAseq analysis clearly showed the marked effect of the lipid emulsion particles on gene expression in macrophages. Interestingly, VLDL-sized emulsion particles reduced the expression of cholesterol biosynthesis genes (GO:0006695), which is consistent with the suppressive effect of unsaturated fatty acids on genes involved in cholesterol synthesis, illustrating the significant impact of the lipid emulsion particles on lipid metabolism in macrophages (Fig 1G) [21,22]. In agreement with previous studies [6,23–25], VLDL-sized emulsion particles also modulated genes involved in the ER stress response (GO:0034976) (Fig 1H) and the inflammatory response (GO:0006954) (Fig 1I).

### LPL is required for the uptake of VLDL-sized emulsion particles by macrophages

To determine whether LPL is required for the uptake of VLDL-sized emulsion particles by macrophages, we treated RAW 264.7 macrophages with heparin, which decreases surface LPL abundance by either releasing LPL from the cell surface or promoting internalization and degradation of LPL [26]. As expected, treatment with heparin markedly reduced macrophage LPL content (Fig 2A). Consistent with a role of LPL in the uptake of VLDL-sized emulsion particles, heparin reduced lipid accumulation in macrophages treated with the emulsion particles (Fig 2B and 2C).

To further assess the role of LPL in macrophage uptake of VLDL-sized emulsion particles, we silenced LPL in human primary macrophages using small interfering RNA (siRNA) (Fig 2D). LPL silencing markedly reduced cellular lipid accumulation in macrophages treated with the emulsion particles, as visualized by BODIPY 493/503 staining (Fig 2E) and supported by flow cytometric analysis (Fig 2F). Consistent with lower lipid uptake, the induction of lipid-sensitive genes *HILPDA*, *PLIN2*, and *PDK4* by the emulsion particles was significantly reduced by LPL silencing (Fig 2G). A similar suppressive effect of LPL silencing on lipid accumulation was observed in macrophages treated with human VLDL (Fig 2H and 2I). Overall, these data indicate that LPL is necessary for macrophage uptake of VLDL.

### Uptake of VLDL-sized emulsion particles by macrophages is dependent on the C-terminal portion of LPL

Based on our data and previous data [6,27,28], we hypothesized that the catalytic function of LPL is essential for macrophage uptake of VLDL. However, GSK264220A, a catalytic inhibitor of LPL, did not significantly alter lipid accumulation in RAW 264.7 cells treated with VLDL-sized emulsion particles [22] (Fig 3A and 3B). Also, the addition of an antibody directed against the catalytic N-terminal portion of human LPL did not noticeably influence lipid

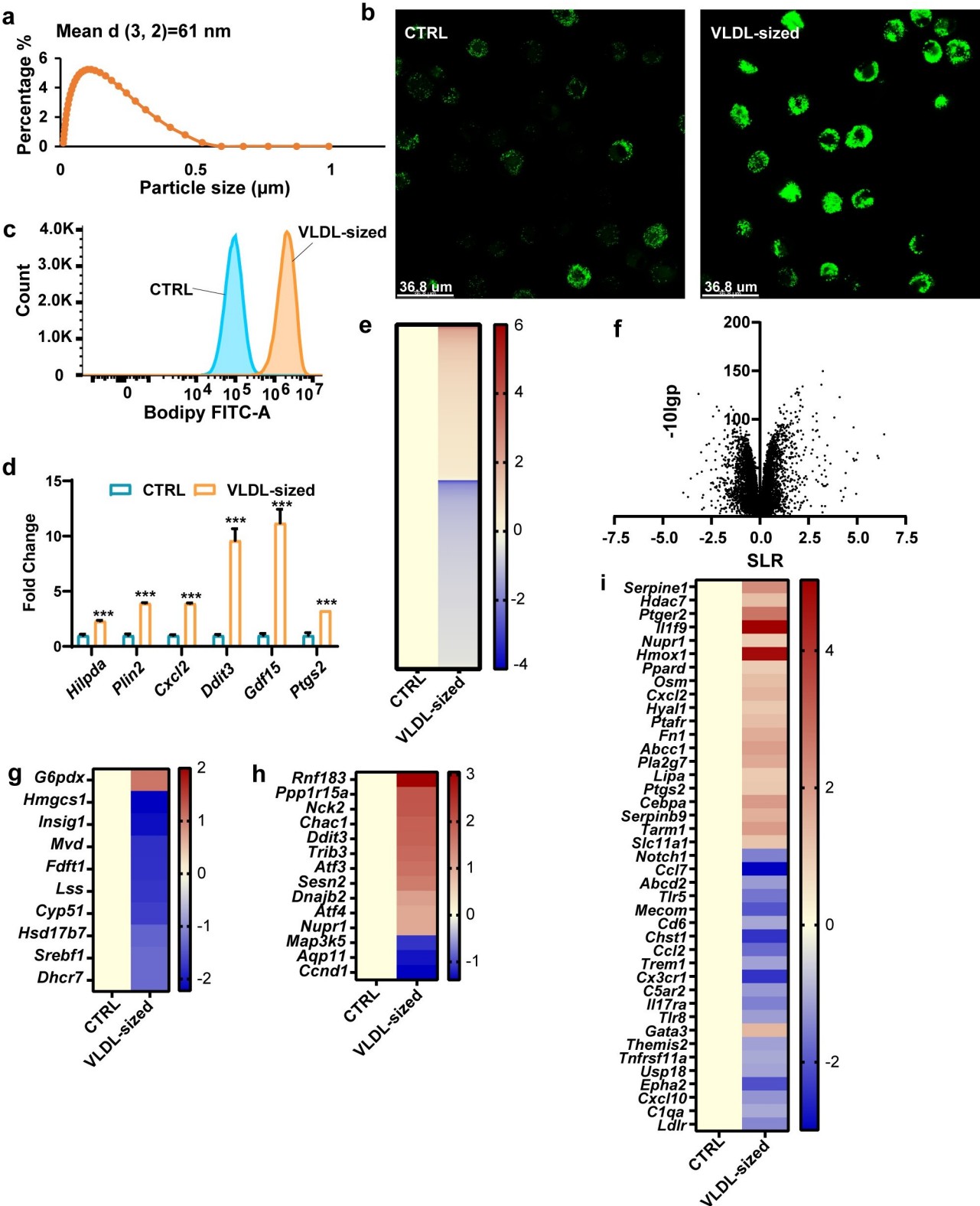

**Fig 1. VLDL-sized emulsion particles promote lipid accumulation in cultured macrophages.** (a) The particle size distribution of VLDL-sized emulsion particles as determined by mastersizer 3000. (b) BODIPY 493/503 staining of intracellular neutral lipids in human primary macrophages treated with 0.5 mM VLDL-sized emulsion particles (referred to as VLDL-sized) for 6 hours (*n* = 6). (c) Mean fluoresce intensity (FITC-A) measured by flow

cytometry of mouse RAW 264.7 macrophages treated with 1 mM VLDL-sized emulsion particles for 6 hours ($n = 3$). (d) mRNA expression of lipotoxic marker genes. (e) Heatmap and (f) volcano plot of RNAseq data of RAW 264.7 macrophages treated with VLDL-sized emulsion particles using all genes with q value less than 0.05 for any comparison. Heatmaps plotted with differentially expressed genes ($p < 0.01$, SLR > 1) involved in cholesterol synthesis (g), the ER stress response (h), and the inflammatory response (i). Bar graphs were plotted as mean ± SD. Scale bar depicts SLR. Data were analysed statistically using Student $t$ test; *$P < 0.05$, **$P < 0.01$, ***$P < 0.001$, ***$P < 0.0001$. (The raw data of RNA-sequencing are available under accession number GSE203250. The FACS data are available under repository ID FR-FCM-Z5K3. Other data can be found in "S1 Raw Data".) ER, endoplasmic reticulum; SLR, signal log ratio; VLDL, very low-density lipoprotein.

accumulation in human monocyte-derived macrophages (Fig 3C and 3D). These data suggest that the uptake of VLDL-sized emulsion particles by macrophages does not require the catalytic function of LPL.

The above finding raises the possibility that LPL may participate in the binding of the VLDL-sized emulsion particles to the macrophage surface. To verify this notion, human primary macrophages were co-treated with VLDL-sized emulsion particles and an antibody (5D2) directed against the C-terminal portion of LPL, which mediates the binding of TRL [23]. Interestingly, the C-terminal hLPL antibody markedly decreased intracellular lipid accumulation, supporting the notion that the C-terminal region of LPL is required for macrophage uptake of VLDL-sized emulsion particles and suggesting that LPL's role in macrophage uptake of VLDL is more as a receptor than as an enzyme (Fig 3E and 3F).

## Macrophage uptake of VLDL-sized emulsion particles is mediated by caveola-mediated endocytosis

Next, we examined whether the uptake of VLDL-sized emulsion particles was mediated by endocytosis. In agreement with this notion, early endosomes could be observed in RAW 264.7 macrophages after loading with VLDL-sized emulsion particles (Fig 4A). To determine if VLDL-sized emulsion particles are taken up by macrophages via clathrin- or caveola-mediated endocytosis, RAW 264.7 macrophages were treated with VLDL-sized emulsion particles in conjunction with the endocytosis inhibitors cholopromazine and genistein, which block clathrin- and caveola-mediated endocytosis, respectively [29]. Whereas cholopromazine showed little to no effect, genistein markedly reduced intracellular lipid accumulation (Fig 4B and 4C), suggesting that VLDL-sized emulsion particles are taken up via caveola-mediated endocytosis. Genistein similarly reduced intracellular lipid accumulation in human primary macrophages treated with the emulsion particles (Fig 4D and 4E).

To further examine the role of caveola in the uptake of VLDL-sized emulsion particles, we silenced the *CAV1* and *CAV2* genes in human primary macrophages using siRNA (Fig 5A and 5B). Caveolins, encoded by *CAV1* and *CAV2*, are the main protein components of caveolae. Silencing of CAV1 and CAV2 markedly decreased intracellular lipid accumulation after loading cells with VLDL-sized emulsion particles, as visualized by confocal microscopy (Fig 5C) and quantified by flow cytometry (Fig 5D). A similar marked reduction in intracellular lipid accumulation upon CAV1 and CAV2 silencing (Fig 5E and 5F) was observed in human primary macrophages treated with human plasma-isolated VLDL. Consistent with a role of LPL, CAV1, and CAV2 in the uptake of VLDL-sized emulsion particles, the change in expression of *IL6* and *MCP1* (*CCL2*) upon treatment with VLDL-sized emulsion particles was altered by siRNA-based silencing of these genes (S1 Fig). Collectively, these results demonstrate that VLDL-sized emulsion particles are taken up via caveolae-mediated endocytosis.

Treatment of RAW 264.7 macrophages with VLDL-sized emulsion particles was associated with marked up-regulation of genes involved in lysosomal function (Fig 6A). Endocytosed lipids destined for degradation are sorted into lysosomes, where they are digested by the enzyme lysosomal acid lipase (LAL). Strikingly, co-treatment of human primary macrophages treated

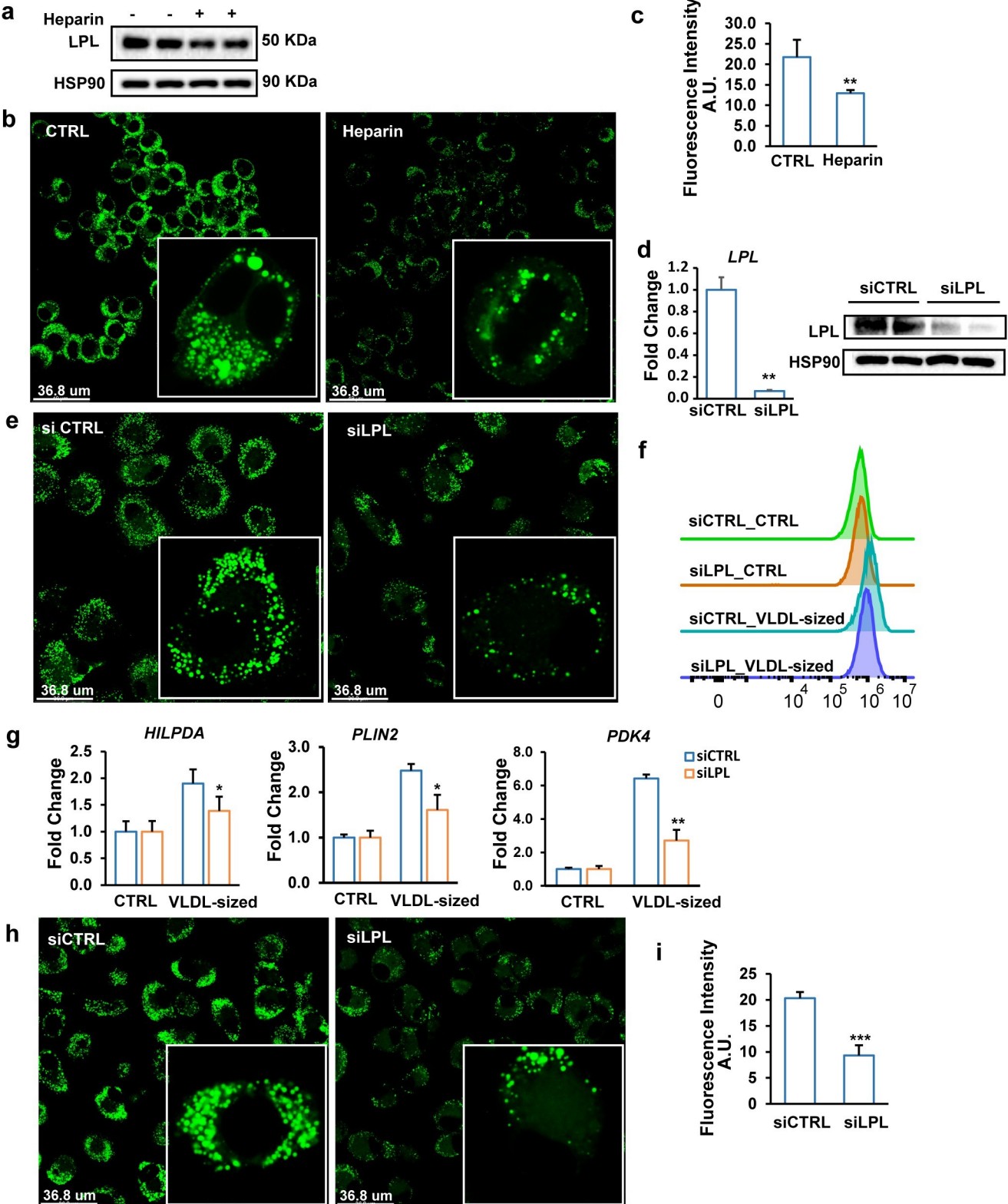

**Fig 2. LPL mediates VLDL uptake in cultured macrophages.** (a) LPL protein levels in RAW 264.7 macrophages treated with 50 UI/ml heparin for 2 hours. (b) BODIPY 493/503 staining of intracellular neutral lipids in RAW 264.7 macrophages treated with 1 mM VLDL-sized emulsion particles for 6 hours in the presence or absence of 50 UI/ml human heparin ($n = 6$). (c) Quantification of the fluorescence images by ImageJ ($n = 6$). (d) LPL mRNA and

protein levels in human primary macrophages treated with control siRNA or LPL siRNA for 48 hours. (e) BODIPY 493/503 staining of intracellular neutral lipids in human macrophages treated with siCTRL or siLPL for 48 hours followed by treatment with 0.5 mM VLDL-sized emulsion particles for 6 hours. (f) Mean fluorescence intensity quantified by flow cytometry ($n = 3$). (g) mRNA expression of selected lipid-sensitive genes. (h) BODIPY 493/503 staining of intracellular neutral lipids in human macrophages treated with siCTRL or siLPL for 48 hours followed by treatment with 0.5 mM human plasma-isolated VLDL for 6 hours ($n = 6$). (i) Mean fluorescence intensity quantified by Image J ($n \geq 4$). The bar graphs were plotted as mean ± SD. Asterisk indicates significantly different from control according to Student $t$ test. $^*p < 0.05$, $^{**}p < 0.01$, $^{***}p < 0.001$. (The FACS data are available under repository ID FR-FCM-Z5KY. Other data can be found in "S1 Raw Data".) LPL, lipoprotein lipase; siRNA, small interfering RNA; VLDL, very low-density lipoprotein.

with VLDL-sized emulsion particles with the LAL inhibitor Lalistat 2 markedly enhanced intracellular lipid accumulation (Fig 6B and 6C). Co-staining of lipids and lysosomes showed that the lipids appeared to be trapped in the lysosomal compartment. Minimal overlap in lipid and lysosomal staining was observed in the absence of LAL inhibition (Figs 6D and S2). Interestingly, non-esterified fatty acids were detected in the culture medium of macrophages previously loaded with VLDL-sized emulsion particles, which was attenuated by LAL inhibition (Fig 6E). This suggests that macrophages can release non-esterified fatty acids after uptake of TG and that this release is dependent on LAL. The increase in lipid accumulation by Lalistat 2 was accompanied by reduced expression of the lipid-sensitive genes *HILPDA*, *PDK4*, *PLIN2*, *CXCL2* (Fig 6F). These data suggest that when LAL is inhibited, endocytosed lipids cannot further be processed and accumulate in lysosomes, preventing the transcriptional activation of lipid-sensitive genes. Taken together, we show that VLDL-sized emulsion particles are degraded via LAL in the lysosome. While most of the liberated fatty acids are destined for storage, a portion is released into the extracellular environment.

## NPC1 silencing retains lipids derived from VLDL-sized emulsion particles in the lysosomes and reduces the extracellular release of free fatty acids

Since NPC1 mediates the lysosomal export of cholesterol, we considered the possibility that NPC1 might also be involved in lysosomal export of VLDL-derived fatty acids. Interestingly, the expression of NPC1 was significantly up-regulated by VLDL-sized emulsion particles in human primary macrophages (Fig 7A). Co-treatment of human primary macrophages treated with VLDL-sized emulsion particles with a chemical inhibitor of NPC1 was associated with markedly elevated intracellular lipid accumulation (Fig 7B), as well as with more pronounced lysosomal staining (Fig 7C). To verify the notion that lipids may be partly retained in the lysosomes upon NPC1 inactivation, we silenced NPC1 in human primary macrophages treated with VLDL-sized emulsion particles and performed co-staining for lipids and lysosomes. Remarkably, NPC1 silencing (Fig 7D) markedly increased the overlap in lipid and lysosomal staining (Figs 7E and S3), suggesting that NPC1 deficiency causes lipids to be retained in lysosomes. Concurrent with the retention of lipids in the lysosomes, NPC1 silencing significantly decreased the levels of non-esterified fatty acids in the medium of macrophages treated with VLDL-sized emulsion particles (Fig 7F). Similarly, chemical inhibition of NPC1 significantly blunted the increase in non-esterified fatty acids in the medium of macrophages treated with the emulsion particles (Fig 7G), concurrent with an increase of lysosome dysfunction-related genes (Fig 7H). Although these data do not necessarily indicate that NPC1 directly mediates free fatty acid efflux from lysosomes, they do suggest that NPC1 is involved in promoting the extracellular release of fatty acids from macrophages.

## STARD3 is required for the lysosomal export of fatty acids derived from VLDL-sized emulsion particles

In addition to NPC1, the expression of *STARD3* was up-regulated by VLDL-sized emulsion particles in human macrophages (Fig 8A). Inasmuch as STARD3 mediates the transport of

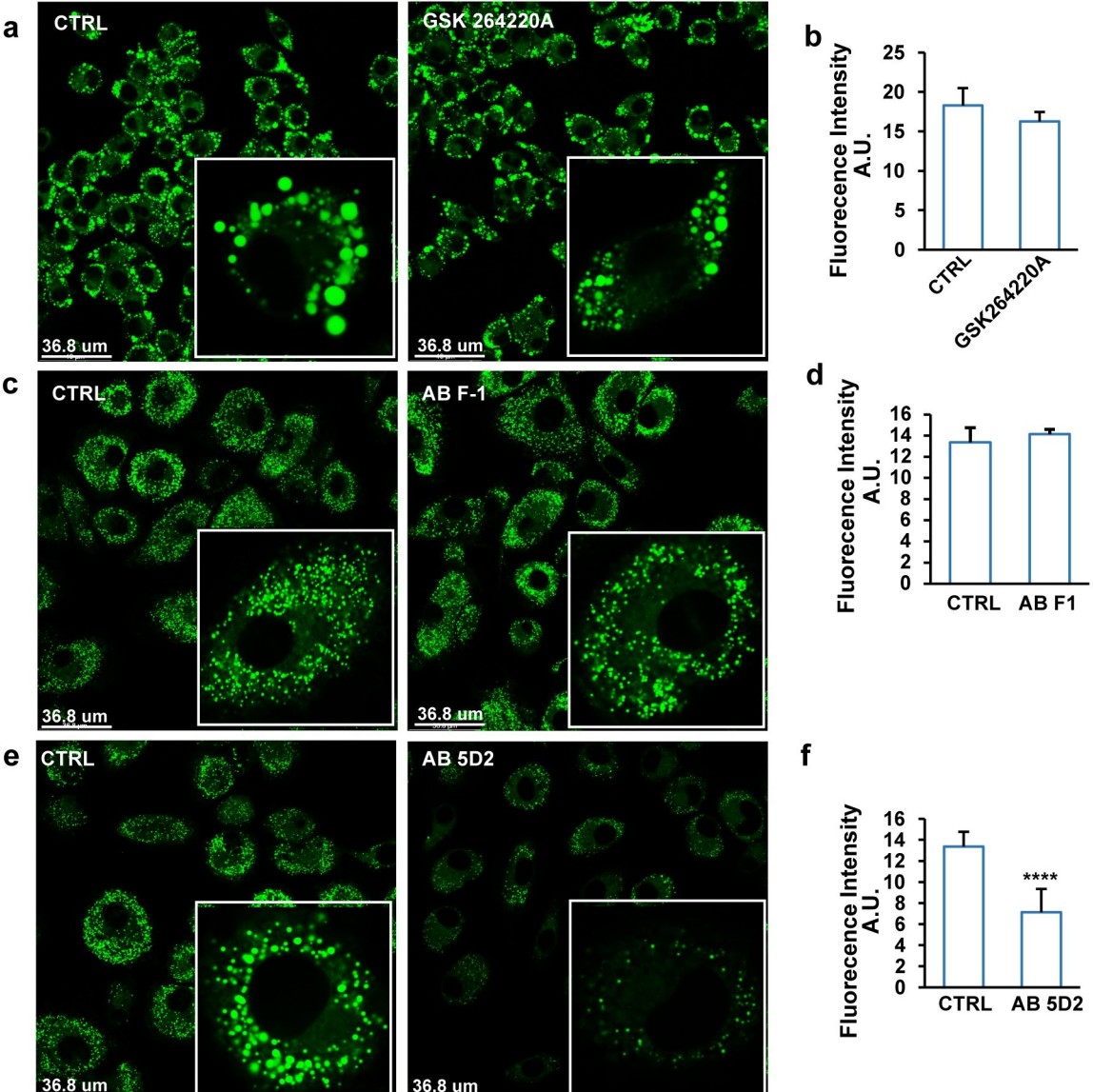

**Fig 3. The C-terminal portion of LPL mediates uptake of VLDL-sized emulsion particles in cultured macrophages.** (a) BODIPY 493/503 staining of RAW 264.7 macrophages treated with 1 mM VLDL-sized emulsion particles for 6 hours in the presence or absence of 0.2 μM of the catalytic LPL inhibitor GSK264220 ($n = 6$). (b) Mean fluorescence intensity quantified by Image J ($n = 4$). (c) BODIPY 493/503 staining of human primary macrophages treated with 0.5 mM VLDL-sized emulsion particles for 6 hours in the presence or absence of antibody F1 targeting the N-terminal portion of LPL (2 μg/ml) ($n = 6$). (d) Mean fluorescence intensity quantified by Image J ($n \geq 4$). (e) BODIPY 493/503 staining of human primary macrophages treated with 0.5 mM VLDL-sized emulsion particles for 6 hours in the presence or absence of antibody 5D2 targeting the C-terminal of LPL (2 μg/ml) ($n = 6$). (f) Mean fluorescence intensity quantified by Image J ($n \geq 4$). The bar graphs were plotted as mean ± SD. Asterisk indicates significantly different from control according to Student $t$ test. ****$p < 0.0001$. (The raw data of bar graphs can be found in "S1 Raw Data".) LPL, lipoprotein lipase; VLDL, very low-density lipoprotein.

cholesterol from lysosomes to the ER [19], we hypothesized that STARD3 may have a similar role in the intracellular trafficking of fatty acids derived from VLDL-sized emulsion particles. In line with this hypothesis, silencing of STARD3 (Fig 8B) in human macrophages treated with VLDL-sized emulsion particles increased lipid accumulation in lysosomes and reduced lipid content in the ER (Figs 8C, 8D, S4, and S5). Unlike silencing of NPC1, silencing of

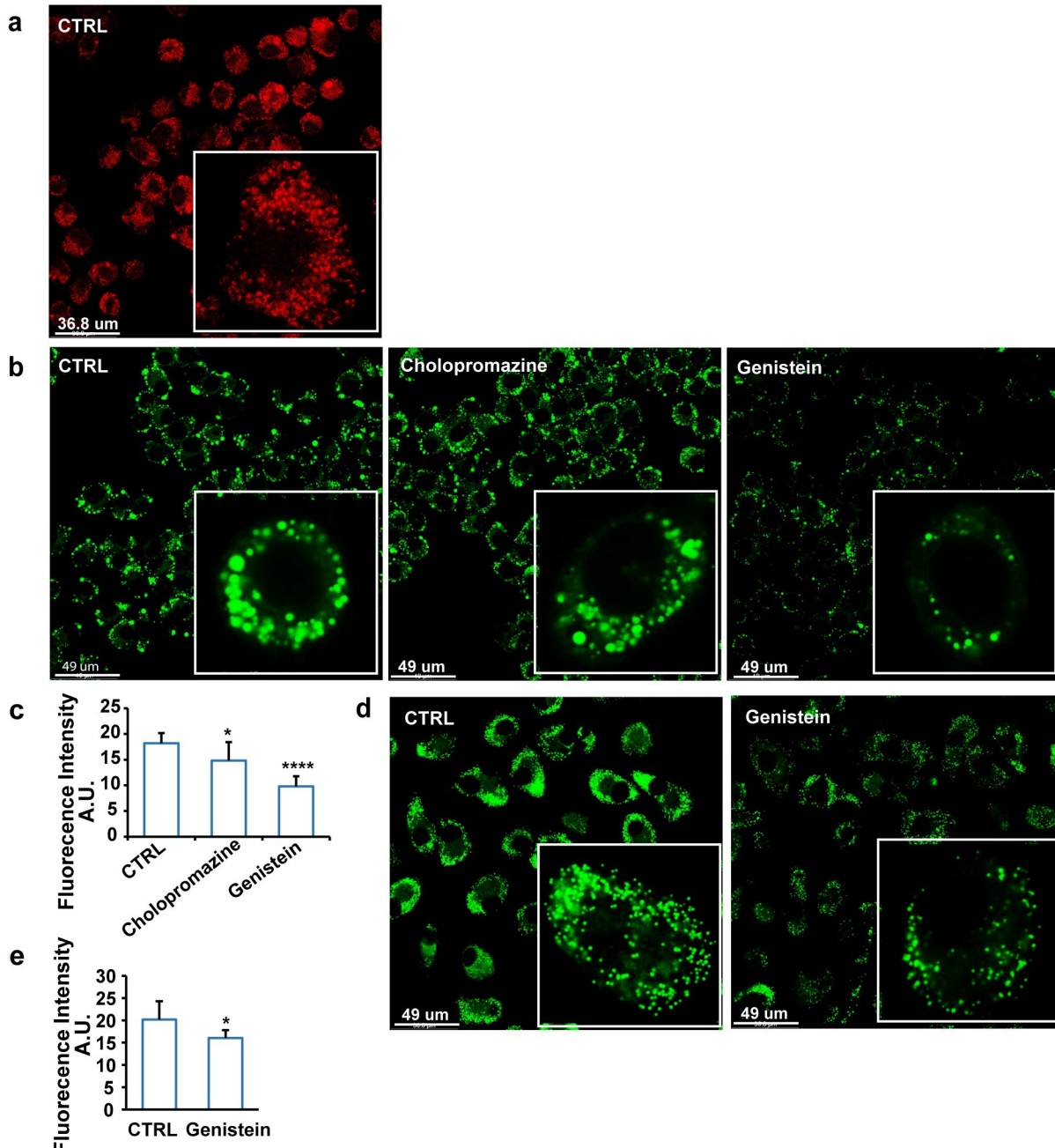

**Fig 4. VLDL-sized emulsion particles are taken up by macrophages via caveola-mediated endocytosis.** (a) Early endosome staining of RAW 264.7 macrophages treated with 1 mM of VLDL-sized emulsion particles for 6 hours ($n = 6$). (b) BODIPY 493/503 staining of RAW 264.7 macrophages treated with 1 mM VLDL-sized emulsion particles for 6 hours in the presence or absence of 10 μg/ml chlorpromazine or 200 μM genistein ($n = 6$). (c) Mean fluorescence intensity quantified by Image J ($n = 6$). (d) BODIPY 493/503 staining of human primary macrophages treated with 0.5 mM VLDL-sized emulsion particles for 6 hours in the presence or absence of 200 μM genistein ($n = 6$). (e) Mean fluorescence intensity quantified by Image J ($n \geq 3$). The bar graphs were plotted as mean ± SD. Asterisk indicates significantly different from control according to Student $t$ test. $^*p < 0.05$, $^{****}p < 0.0001$. (The raw data of bar graphs can be found in "S1 Raw Data".) VLDL, very low-density lipoprotein.

STARD3 did not increase intracellular lipid accumulation nor did it influence the efflux of fatty acids into the medium (Fig 8E). These findings are most consistent with a scenario in

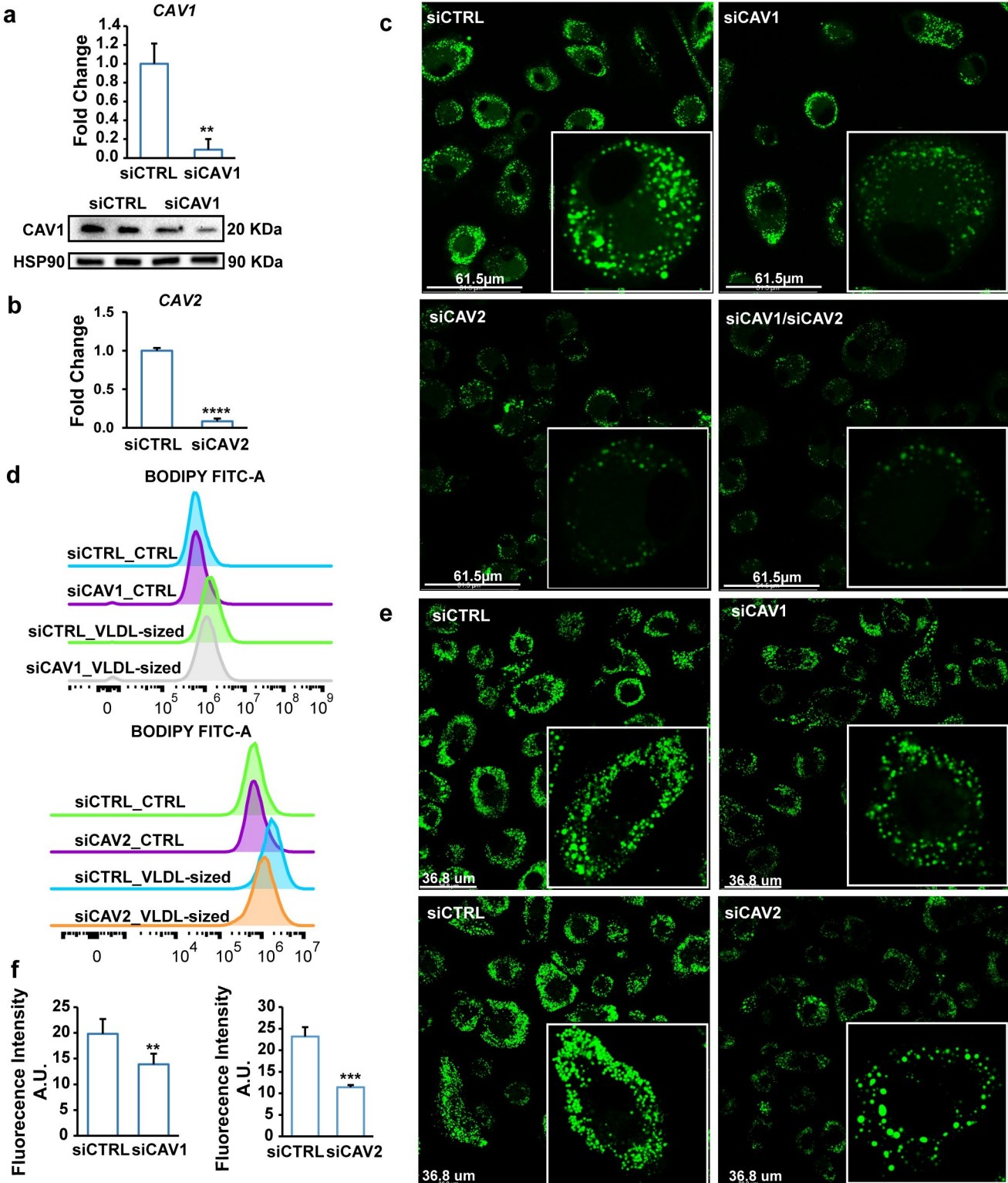

**Fig 5. Silencing of Caveolin 1 and 2 impairs uptake of VLDL by macrophages.** (a) mRNA and protein levels of Caveolin 1 after 48-hour treatment with siCTRL or siCAV1. (b) mRNA expression of Caveolin 2 after 48-hour treatment with siCTRL or siCAV2. (c) BODIPY 493/503 staining of human macrophages treated with siCTRL, siCAV1, or siCAV2 for 48 hours followed by treatment with 0.5 mM VLDL-sized emulsion particles for 6 hours ($n = 6$). (d) Mean fluorescence intensity quantified by flow cytometry ($n = 3$). (e) BODIPY 493/503 staining of human macrophages treated with siCTRL, siCAV1, or siCAV2 for 48 hours followed by treatment with 0.5 mM human plasma-isolated VLDL for 6 hours ($n = 6$). (f) Mean fluorescence intensity quantified by

Image J ($n \geq 4$). The bar graphs were plotted as mean ± SD. Asterisk indicates significantly different from control according to Student $t$ test. **$p < 0.01$, ***$p < 0.001$. (The FACS data are available under repository FR-FCM-Z5JV and FR-FCM-Z5KZ. Other data can be found in "S1 Raw Data".) VLDL, very low-density lipoprotein.

which STARD3 is involved in commuting fatty acids from the lysosome to the ER but does not contribute to the extracellular export of fatty acids. Taken together, our data indicate that NPC1 and STARD3 have differential roles in the transport of VLDL-derived fatty acids from

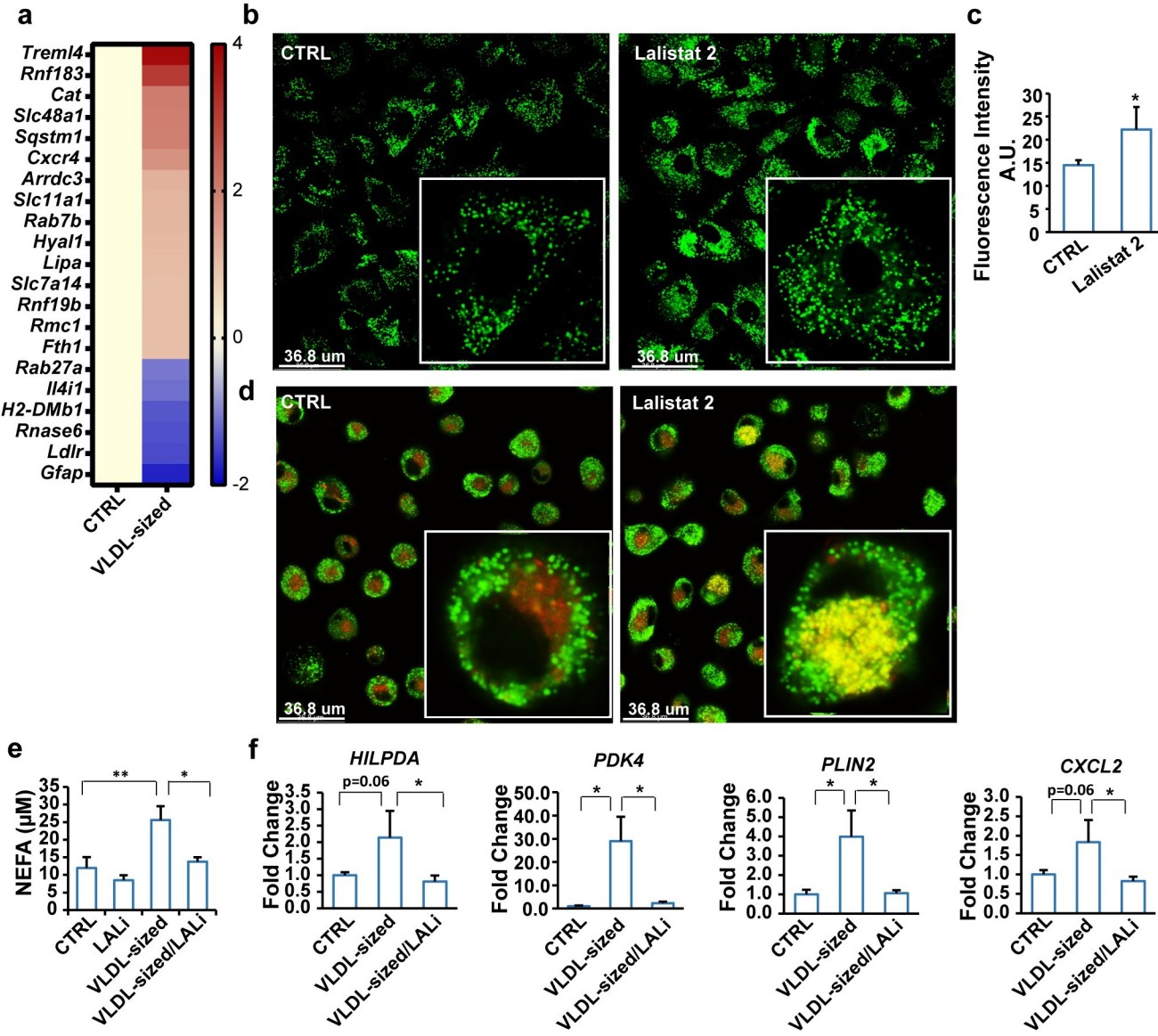

**Fig 6. VLDL-sized emulsion particles are degraded by LAL.** (a) Heatmap showing changes in expression of genes involved in lysosome activity in RAW 264.7 macrophages treated with 1 mM VLDL-sized emulsion particles for 6 hours ($p < 0.01$, SLR > 1). Scale bar depicts SLR. (b) BODIPY 493/503 staining of human macrophages treated with 0.5 mM VLDL-sized emulsion particles for 6 hours in the presence or absence of 30 μM Lalistat 2 ($n = 6$). The cells were washed twice with PBS and then incubated in fresh medium without VLDL-sized emulsion particles for another 24 hours. (c) Mean fluorescence intensity quantified by Image J ($n = 4$). (d) Co-staining of lysosome (red) and neutral lipids (green) in human macrophages ($n = 6$). (e) Non-esterified fatty acid levels in the medium. (f) mRNA levels of lipid-sensitive genes in human macrophages. The bar graphs were plotted as mean ± SD. Asterisk indicates significantly different compared with control or in marked comparisons according to Student $t$ test. *$p < 0.05$, **$p < 0.01$. (The raw data of RNA-sequencing are available under accession number GSE203250. Other data can be found in "S1 Raw Data".) LAL, lysosomal acid lipase; SLR, signal log ratio; VLDL, very low-density lipoprotein.

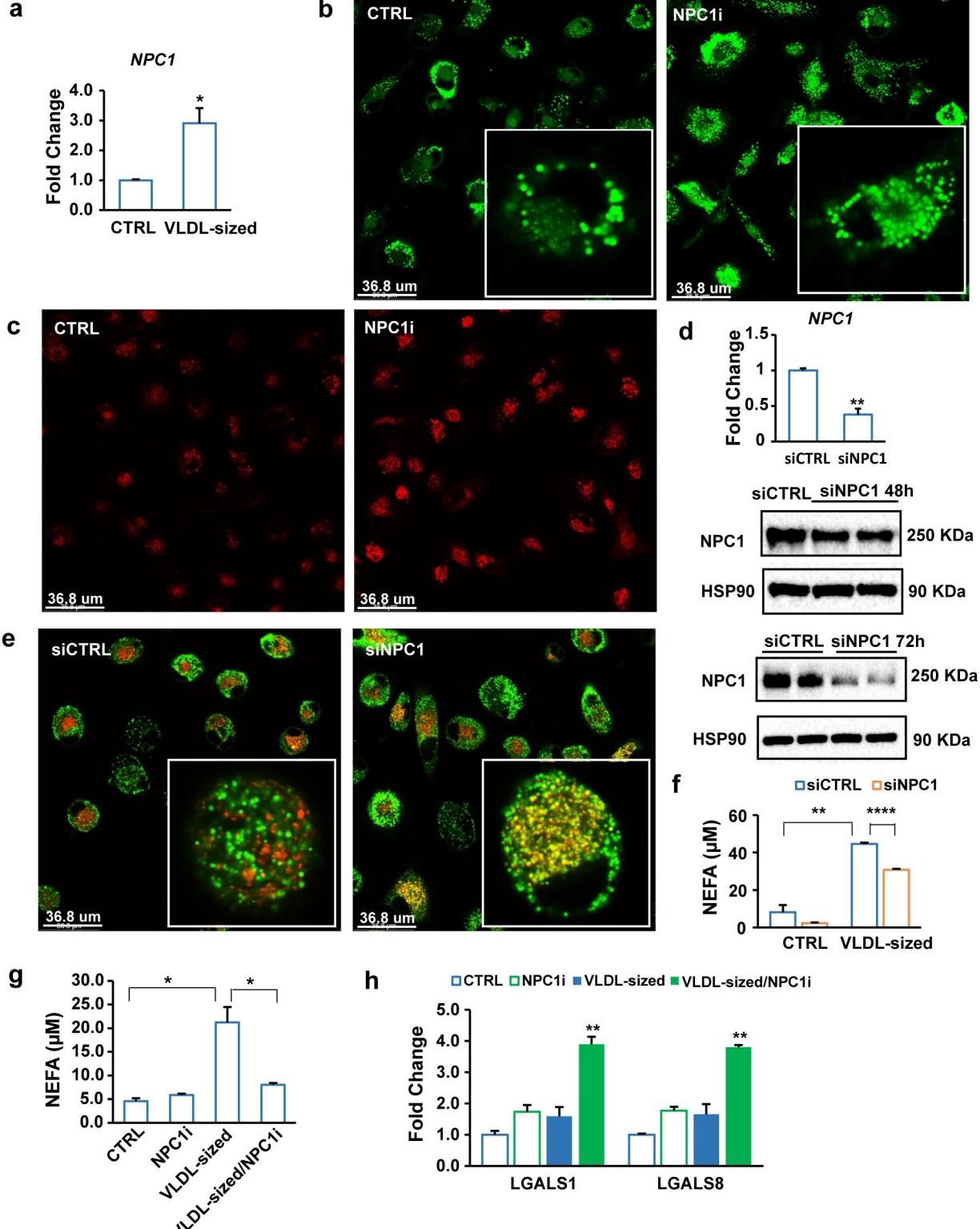

**Fig 7. Part of the internalized TG in VLDL-sized emulsion particles is released from macrophages after lysosomal processing.** (a) mRNA levels of NPC1 in human macrophages treated with 1 mM VLDL-sized emulsion particles for 6 hours. Cells were washed twice with PBS and then incubated in fresh medium for 16 hours. (b) BODIPY 493/503 staining or (c) lysosomal staining of human macrophages treated with 0.5 mM VLDL-sized emulsion particles for 6 hours in the presence or absence of 5 μM NPC1 inhibitor U18666A (*n* = 6). (d) mRNA and protein

levels of NPC1 in human macrophages treated with siCTRL or siNPC1. (e) Co-staining of lysosome (red) and neutral lipids (BODIPY 493/503, green) in human macrophages treated with siCTRL or siNPC1 for 72 hours followed by treatment with 0.5 mM VLDL-sized emulsion particles for 24 hours ($n = 6$). Thereafter, the cells were washed twice with PBS and incubated in fresh medium without VLDL-sized emulsion particles for 24 hours. (f) Non-esterified fatty acid concentration in culture medium from human macrophages treated with siCTRL or siNPC1 for 48 hours followed by treatment with 0.5 mM VLDL-sized emulsion particles for 6 hours or (g) treatment with 0.5 mM VLDL-sized emulsion particles for 6 hours in the presence or absence of 5 μM NPC1 inhibitor U18666A. After the treatments, the cells were washed twice with PBS, followed by incubation in fresh medium without VLDL-sized emulsion particles for 16 hours. (h) mRNA levels of galectin-encoding genes. The bar graphs were plotted as mean ± SD. Asterisk indicates significantly different compared with control or in marked comparisons according to Student $t$ test. $^*p < 0.05$, $^{**}p < 0.01$, $^{****}p < 0.0001$. (The raw data of bar graphs can be found in "S1 Raw Data".) TG, triglyceride; VLDL, very low-density lipoprotein.

the lysosome to other (extra)cellular compartments. Specifically, NPC1 somehow promotes the extracellular release of (lysosomal) fatty acids, while STARD3 is involved in the transfer of lysosomal fatty acids to the ER for subsequent storage as TG.

## Macrophages take up chylomicron-sized emulsion particles via a similar mechanism as VLDL-sized emulsion particles

In contrast to VLDL, chylomicrons are not able to pass into the intima. Nevertheless, chylomicrons may come into direct contact with macrophages in the mesenteric lymph nodes. In addition, severely elevated chylomicron levels are associated with the accumulation of chylomicron-derived lipids within skin macrophages, giving rise to eruptive xanthomas. Accordingly, we asked whether the mechanism of uptake of chylomicrons by macrophages resembles the mechanism of uptake of VLDL. To that end, we repeated the studies using lipid emulsion particles with a mean diameter of 240 nm (S6a Fig), referred to as chylomicron-sized emulsion particles, in agreement with the reported particle sizes of median-sized chylomicrons [1]. Treatment of RAW 264.7 macrophages and human primary macrophages with these particles for 6 hours led to marked lipid accumulation (S6b Fig) and increased expression of lipid-sensitive genes (S6c Fig). Similar to VLDL-sized emulsion particles, we found that LPL, but not its catalytic activity, is required for the uptake of CHYL-sized emulsion particles by macrophages (S7 and S8 Figs) and that this process is driven by caveolae-mediated endocytosis (S9 Fig), involving caveolin 1 and 2 (S10 Fig). Additionally, the intracellular processing of CHYL-sized emulsion particles is mediated by LAL (S11 Fig). Taken together, these data indicate that the mechanisms of macrophage uptake of VLDL- and CHYL-sized emulsion particles are highly similar.

## Discussion

By virtue of its ability to enter the intima and be taken up by vascular macrophages, VLDL may contribute to atherosclerosis. Here, using artificial VLDL-sized emulsion particles and human VLDL, we studied the mechanism of uptake of VLDL particles by macrophages. We found that macrophage uptake of VLDL requires LPL and is mediated by the lipoprotein-binding C-terminal domain of LPL and not by the catalytic N-terminal domain. Subsequent internalization of VLDL-TG by macrophages was shown to occur via caveolae-mediated endocytosis, followed by TG hydrolysis by LAL in the lysosome. Intriguingly, NPC1 was found to promote the extracellular efflux of fatty acids from lysosomes, while STARD3 is involved in the transfer of lysosomal fatty acids to the ER for subsequent storage as TG. These data suggest that macrophages have the remarkable capacity to excrete part of the internalized TG as fatty acids. Our data elaborate on the model put forward by Lindqvist and colleagues many years ago, who on the basis of studies using radiolabeled and unlabeled human VLDL proposed that incubation of macrophages with VLDL leads to TG accumulation via uptake of intact VLDL,

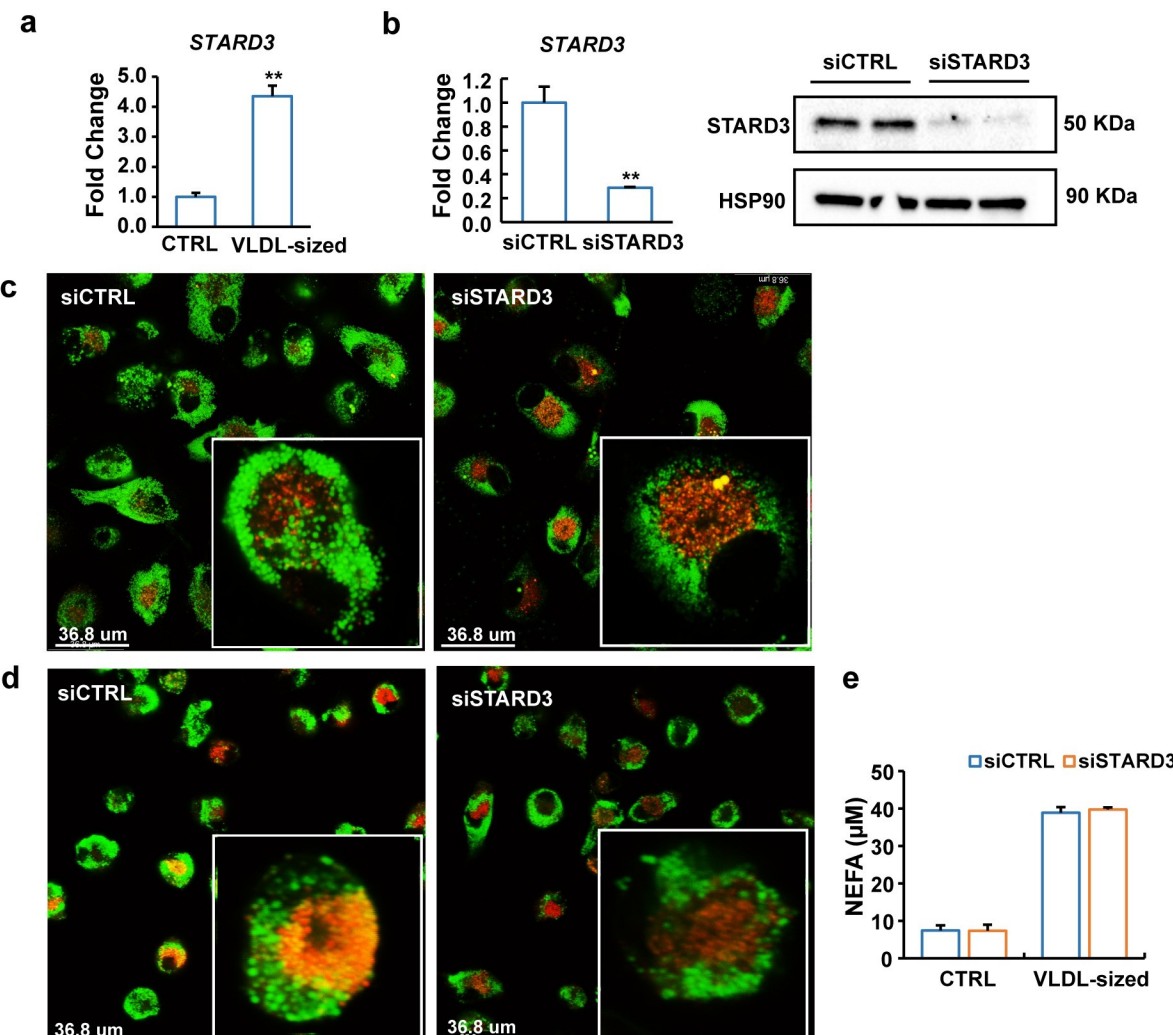

**Fig 8. NPC1 and STARD3 have differential roles in the processing of VLDL-sized emulsion particles.** (a) mRNA levels of STARD3 in human macrophages after treatment of 1 mM VLDL-sized emulsion particles for 6 hours. Cells were washed twice with PBS and then incubated in fresh medium for 16 hours. (b) mRNA and protein levels of STARD3 in human macrophages treated with siCTRL or siSTARD3 for 24 hours. (c) Co-staining of lysosomes (red) with neutral lipids (BODIPY 493/503, green) in human macrophages treated with siCTRL or siSTARD3 for 48 hours followed by treatment with 0.5 mM VLDL-sized emulsion particles for 24 hours ($n = 6$). Thereafter, the cells were washed twice with PBS and incubated in fresh medium without VLDL-sized emulsion particles for 24 hours. (d) Co-staining of ER (red) and neutral lipids (BODIPY 493/503, green) in human macrophages treated with siCTRL or siSTARD3. (e) Non-esterified fatty acid concentration in culture medium from siCTRL or siSTARD3-treated human macrophages. The cells were washed twice with PBS and incubated in fresh medium without VLDL-sized emulsion particles for 16 hours. The bar graphs were plotted as mean ± SD. Asterisk indicates significantly different from control according to Student $t$ test. $^{**}p < 0.01$. (The raw data of bar graphs can be found in "S1 Raw Data".) ER, endoplasmic reticulum; VLDL, very low-density lipoprotein.

mediated by either a receptor or a nonreceptor-mediated pathway and involving phagocytosis or endocytosis [30].

LPL is mainly known for its ability to catalyze the hydrolysis of TG in VLDL and chylomicrons and thereby promote the uptake of plasma TG-derived fatty acids in tissues such as the heart, skeletal muscle, white adipose tissue, and brown adipose tissue [1,31]. In addition to catalyzing TG hydrolysis, LPL can facilitate the binding and uptake of lipoprotein particles by cells independent of lipolysis [32,33]. Through its ability to interact with lipoproteins on the one hand, and heparan sulfate proteoglycans or specific surface receptors on the other hand,

LPL can function as a bridge between lipoproteins and the cell surface [34]. For example, in macrophages—which are characterized by high levels of LPL expression—LPL was found to enhance the uptake of oxLDL [35]. Previous studies have shown that LPL promotes the uptake TRL-TG and -cholesterol by macrophages in vitro [30,36]. However, thus far it was not fully clear whether the stimulation of TRL-TG uptake by LPL is mainly mediated by the abovementioned lipolysis-independent bridging function—leading to whole particle uptake—or requires actual TG hydrolysis followed by cellular uptake of fatty acids [37]. Using antibodies directed against the N- or C-terminal domain of LPL, we found that macrophage uptake of TG contained in VLDL-sized emulsion particles is dependent on particle binding by LPL but not on the catalytic function of LPL. Whether binding to HSPG-bound LPL is sufficient to trigger endocytosis of the VLDL-like particles or requires an additional receptor remains to be determined.

Previous evidence suggested that LPL-catalyzed lipolysis is not required for the uptake of VLDL-TG by cultured macrophages but does have a modest stimulatory effect [30]. Further work is necessary to better define the role of the catalytic function of LPL in macrophage lipid metabolism. For example, it would be of interest to know the impact of the replacement of wild-type LPL by catalytically inactive mutated LPL on lipid uptake and metabolism in macrophages treated with VLDL-sized emulsion particles [38]. As an alternative, we considered the use of the LPL inhibitor orlistat. However, a major drawback of orlistat is that it also inhibits LAL [39], which would seriously complicate the interpretation of the data.

Our data suggest that macrophages internalize entire VLDL particles via endocytosis. Uptake of entire TRL particles has been previously observed in vascular endothelial cells of BAT and WAT [40–42]. In endothelial cells, the TRL are routed through the endosomal-lysosomal pathway, where they undergo LAL-mediated processing [43]. This route strongly resembles the pathway we observed for VLDL-sized emulsion particles in human primary macrophages.

It is well established that LDL and oxLDL are taken up by cells via receptor-mediated clathrin-dependent endocytosis [44–49]. In contrast, our data indicate that VLDL is taken up by macrophages via caveolae-mediated endocytosis. Specifically, we observed that genistein markedly reduced lipid accumulation in macrophages treated with VLDL-sized emulsion particles, whereas cholopromazine had no effect. Gene silencing of *CAV1* and especially *CAV2* also markedly reduced lipid accumulation in VLDL-treated macrophages. The difference in the type of endocytosis mediating the uptake of LDL and VLDL might be explained by the different sizes of the 2 types of particles or by differences in the composition of the lipid cargo [50]. Interestingly, previous studies found that CAV1 is enriched in lipid droplet fractions of endothelial cells and adipocytes [51,52]. It can be hypothesized that in macrophages, part of the CAV1 is similarly associated with lipid droplets, which might imply that the role of caveolins in lipid processing in macrophages may go beyond endocytosis.

The internalization and subsequent breakdown of pathogens, apoptotic cells, or particles such as lipoproteins burden the macrophage with potentially toxic macromolecules that must either be metabolized or expelled [53]. For example, internalized cholesterol can either be esterified to fatty acids and stored or it can be exported out of the macrophages via ABCA1 to apolipoprotein A-I to produce precursors for HDL particles. Through these mechanisms, macrophages are able to limit the toxic effects of excessive free cholesterol levels on the cell membrane. Similar to cholesterol, elevated intracellular levels of free fatty acids can also be damaging to cells. To restrict this lipotoxicity, macrophages and other cells are able to convert fatty acids into TG as well as use the fatty acids as fuel. However, in contrast to cholesterol, there is very little evidence in the literature that macrophages export fatty acids. Lindqvist and colleagues showed that the presence of bovine serum albumin in the culture medium markedly

decreased TG content in lipid-laden macrophages, raising the suggestion that macrophages are capable of mobilizing its stored TG [30]. Consistent with this notion, we found that macrophages loaded with VLDL-sized emulsion particles release fatty acids into the medium. Moreover, we observed that this process was impaired by inactivation of LAL and NPC1. At this stage, it is unclear if extracellular fatty acid efflux is dependent on the fusion of the lysosomes with the plasma membrane or requires fatty acid transport through the cytoplasm. Since NPC1 was specifically dismissed as fatty acid transporter [54], the decreased fatty acid release upon NPC1 inactivation is probably secondary to impaired lysosomal lipolysis due to lysosomal accumulation of cholesterol, rather than reflecting a direct role of NPC1 as fatty acid transporter. Previously, it was suggested that lipid-laden macrophages may release fatty acids via the action of a nonlysosomal (presumably cytoplasmic) neutral triglyceride lipase [55]. Based on our data, we cannot exclude that the increase in fatty acid release upon NPC1 silencing was dependent on TG storage in lipid droplets and ATGL-mediated lipolysis prior to the release of the fatty acids into the medium.

Similar to our data, Mashek and colleagues recently found that fatty acids liberated by lipophagy in lysosomes can be transported out of hepatocytes and suggested that the effluxed fatty acids may be available for uptake by the same cell [56]. Based on our data, it is impossible to say whether the extracellular efflux of lysosome-derived fatty acids is needed for the storage of VLDL-TG in macrophages via re-uptake of the fatty acids. One could wonder about the rationale for exporting fatty acids from the cell if most of the fatty acids are re-taken up. Rather, the efflux of fatty acids may be a mechanism to rid the macrophage of excess fatty acids acquired via endocytosis and phagocytosis. It can be speculated that the fatty acids released by macrophages may be used as fuel by neighboring cells, similar as has been suggested for endothelial cells [52].

StAR-related lipid transfer domain-3 (STARD3) is a sterol-binding protein that promotes sterol transport by creating ER–endosome contact sites [57]. We found that STARD3 is involved in the transfer of VLDL-derived fatty acids from the lyso/endosome to the ER. Whether STARD3 directly binds fatty acids or stimulates transport by creating membrane contact sites requires further study. Our data suggest that in contrast to NPC1, STARD3 does not promote the extracellular efflux of fatty acids. NPC1 and STARD3 thus have distinct roles in the transport of VLDL-derived fatty acids from the lysosome to other (extra)cellular compartments.

Our key findings on the uptake and intracellular processing of VLDL-sized emulsion particles in macrophages were reproduced when using the larger CHYL-sized emulsion particles, suggesting that TRL are processed through a common mechanism. However, it should be noted that the range in particle sizes of our artificial CHYL-sized emulsion particles is smaller than that of human chylomicrons. Accordingly, it cannot be excluded that human chylomicrons are also taken up via other mechanisms, such as micropinocytosis. Previous in vitro studies already found large similarities in the mechanism of VLDL and chylomicron uptake and degradation by macrophages, showing among other things that macrophages are able to take up intact VLDL and chylomicron particles [30,36]. The in vivo relevance of uptake of chylomicrons by macrophages is likely limited. In contrast to VLDL and its remnants, chylomicrons are not able to penetrate the arterial wall to be taken up by macrophages. One specific location where chylomicrons do get in direct contact with macrophages is in the mesenteric lymph nodes, where excessive uptake of chylomicrons can lead to the formation of giant macrophage foam cells, as observed in ANGPTL4-deficient mice on a high fat diet [58].

Our study also has limitations. First, most of our experiments were conducted with artificial TG-rich emulsion particles. These VLDL-mimicking particles were generated with a microfluidizer using casein as emulsifier. It should be noted, though, that our key findings were

verified with human VLDL. In addition, our data on the role of LPL in VLDL-TG uptake are fully consistent with previous studies that used human VLDL. The similarity in the mechanism of uptake of human VLDL and artificial VLDL-sized emulsion particles containing casein as emulsifier suggests that macrophages recognize TRL particles on the basis of size and/or lipid content rather than a specific protein component. A second limitation of our study is that the TG in VLDL-sized emulsion particles were not radioactively or fluorescently labeled. Consequently, we cannot fully exclude that part of the stored lipids may be endogenously synthesized by the macrophage in response to treatment with VLDL-sized emulsion particles. However, this contribution likely is small. In the future and when available, fluorescently labeled TG could be incorporated into the VLDL-sized emulsion particles to enable direct intracellular visualization of exogenous lipids.

In conclusion, our data suggest that the uptake of TRL-derived TG by macrophages is mediated by the lipid-binding function of LPL. After binding to LPL, TRL-TG are taken up by caveolin-mediated endocytosis, followed by LAL-catalyzed hydrolysis in the lysosomes. Subsequent processing of TRL-derived fatty acids toward storage requires the proteins NPC1, which was found to promote the extracellular efflux of fatty acids from lysosomes, and STARD3, which is involved in the transfer of lysosomal fatty acids to the ER for subsequent storage as TG. Our data provide key new insights into how macrophages take up and process TRL.

## Methods

### Preparation of VLDL- and chylomicron-sized lipid emulsions

The VLDL- and chylomicron-sized lipid emulsions were prepared with commercial sunflower oil (Gwoon, the Netherlands) with a microfluidizer at pressures of 400 bar and 1,200 bar, respectively, for 5 cycles. Sodium caseinate (purity 97%, Excellion, FrieslandCampina, the Netherlands) was used as an emulsifier. Caseinate was solubilized in PBS overnight at 4°C to a final concentration of 1% (w/w). Before making the nano-emulsions, 10% sunflower oil mixed with caseinate solution was pre-emulsified using an IKA Ultra turrax for 1 minute at 10,000 rpm/min. The size of the droplets was measured using a Mastersizer 3000 (Malvern Panalytical, United Kingdom).

### Cell culture

**RAW 264.7 macrophages.** RAW 264.7 macrophages were cultured in DMEM supplemented with 10% FCS and 1% p/s. Cells were seeded at a density of 52,000 cells/cm$^2$ and cultured overnight before treatment with emulsions for 6 hours.

### Human buffy-coat primary macrophages

**PBMC isolation.** Human buffy-coat blood was obtained from Sanquin, the Netherlands. Briefly, 25 ml of buffy-coat blood diluted 1:1 with PBS was added to 50 ml Leucosept tubes containing 15 ml of Ficoll-Paque followed by centrifugation for 15 minutes at 800 RCF at room temperature. Afterwards, the PBMC layer was collected and washed with cold PBS for 3 times. Approximately 70 μM cell strainers were used to remove clumps/fat residues.

**Monocytes isolation.** Moncytes were isolated by MojoSort CD14 positive selection kits (Biolegend, California, United States) and LS columns (Miltenyi Biotec, Bergisch Gladbach, Germany). Briefly, 10 μl CD14 Nanobeads (10× pre-diluted in MACS buffer (PBS + 0.5% BSA + 2 mM EDTA)) and 90 μl MACS buffer were added per $1 \times 10^7$ PBMCs and incubated at 4°C for 15 minutes (gently mixed every 5 minutes). Afterwards, the monocytes were separated by LS columns using a Miltenyi QuadroMACS Separator.

Isolated cells were cultured in RPMI medium supplemented with 10% FCS, 1% GlutaMax, and 1% p/s at a density of $1 \times 10^6$ cells/ml. Approximately 5 ng/ml of granulocyte-macrophage colony-stimulating factor (GM-CSF; Miltenyi) was used to differentiate monocytes into macrophages. After full differentiation, cells were seeded at a density of 100,000 cells per cm$^2$ for siRNA assay and 150,000 cells per cm$^2$ for the other assays.

## Inhibitors and heparin assay

Cells were pre-incubated with chemical inhibitors (30 μM of Lalistat 2, 0.2 μM of GSK264220A, 200 μM of genistein, 10 μg/ml of chlorpromazine, 5 μM of U18666A) for 1 hour and then continuously treated with lipid emulsions for 6 hours.

For the heparin assay, cells were treated with 50 UI/ml of heparin for 2 hours, followed by 2 times washing with PBS. Emulsions were subsequently added to the cells together with the same concentration of heparin.

Human LPL F:1 (Santa Cruz Biotechnology, California, USA) and 5D2 antibodies were added to the cell culture medium 2 hours prior to emulsion loading at the concentration of 2 μg/ml (1:100 dilution).

Genistein, Lalistat 2, chlorpromazine, and heparin were obtained from Sigma-Aldrich, Missouri, USA. GSK264220A was from Tocris (bio-techne), Abingdon, UK. LPL 5D2 antibody was contributed by Dr. Anne Beigneux, Department of Medicine, David Geffen School of Medicine, UCLA, USA.

## Free fatty acids assay

For functional study of NPC1 and STARD3, GM-CSF-derived macrophages were loaded with VLDL-sized emulsion particles. After washing the cells twice with PBS, the medium was refreshed and cells were left for 16 hours. Then, medium was collected for assessment of free fatty acids using the free fatty acids kit (Instruchemie, the Netherlands) following the manufacturer's instructions.

## Quantitive RT-PCR

Total RNA was isolated using TRizol Reagent (Thermo Fisher Scientific, Massachusetts, USA). cDNA was synthesized using iScript cDNA Sythesis Kit (Bio-Rad Laboratories, California, USA) following the manufacturer's protocol. Real-time polymerase chain reaction (RT-PCR) was performed on the CFX 384 Touch Real-Time detection system (Bio-Rad Laboratories, California, USA), using the SensiMix (BioLine, London, UK) protocol for SYBR green reactions. Mouse/human 36B4 expression was used for normalization.

## Immunoblotting

The cell lysates were prepared using RIPA Lysis and Extraction Buffer (Thermo Fisher Scientific, Massachusetts, USA) or with self-prepared NP40 lysis buffer (50 mM Tris-HCl (pH 8.0), 0.5% NP40, 150 mM NaCl, 5 mM MgCl$_2$) for cell membrane-binding proteins (LPL, Caveolin 1) and quantified with Pierce BCA Protein Assay Kit Buffer (Thermo Fisher Scientific, Massachusetts, USA). The cell lysates were separated by electrophoresis on pre-cast 4% to 15% polyacrylamide gels and transferred onto nitrocellulose membranes using a Trans-Blot Semi-Dry transfer cell (Bio-Rad Laboratories, California, USA), blocked in 5% skim milk in TBS-T (TBS buffer supplied with 1‰ TWEEN 20) and incubated with LPL antibody (F:1), caveolin-1 antibody (4H312), STARD3 antibody (H-1) (sc-166215) (Santa Cruz Biotechnology, Texas, USA), and NPC1 antibody (ab134113) (Abcam, Cambridge, UK) overnight at 4°C. Secondary

antibody incubation was performed at room temperature for 1 hour. HSP90 was used for normalization (antibody was purchased from Cell Signaling Technology, Massachusetts, USA). Images were gained using the ChemiDoc MP system (Bio-Rad Laboratories, California, USA).

## RNAseq analysis

Cells were treated with 1 mM VLDL-sized or chylomicron-sized lipid emulsions for 6 hours and harvested for total RNA isolation. The experiments were performed in both biological and technical triplicates. Samples of each condition from 1 experiment were pooled. Transcriptome analysis by RNA-sequencing was performed by BGI Hong Kong Company Limited (Hong Kong) following a standard protocol. In brief, RNA samples were prepared using RNeasy Mini Kit (Qiagen, Hilden, Germany) following the manufacturer's instructions. Samples were then shipped to BGI for library construction and RNA sequencing runs on the BGI-SEQ-500 platform [59]. Genomic DNA was removed with 2 digestions using Amplification grade DNAse I (Invitrogen, USA). The RNA was sheared and reverse transcribed using random primers to obtain cDNA, which was used for library construction. The library quality was determined by using Bioanalyzer 2100. Thereafter, the library was used for 150-bp paired-end sequencing on the BGISEQ-500 sequencing platform. All the generated raw sequencing reads were filtered, by removing reads with adaptors, reads with more than 10% of unknown bases, and low-quality reads. Clean reads were then obtained and stored as FASTQ format.

The RNA-seq reads were used to quantify transcript abundances. To this end, the tool Salmon (version 0.14.1) [60] was used to map the reads to the GRCm38.p6 mouse genome assembly-based transcriptome sequences as annotated by the GENCODE consortium (release M22) [61]. The obtained transcript abundance estimates and lengths were then imported in R using the package tximport, scaled by average transcript length and library size, and summarized on the gene level. Such scaling corrects for bias due to correlation across samples and transcript length and has been reported to improve the accuracy of differential gene expression analysis [62]. Differential gene expression was determined using the package limma [63] utilizing the obtained scaled gene-level counts. Briefly, before statistical analyses, nonspecific filtering of the count table was performed to increase detection power, based on the requirement that a gene should have an expression level greater than 10 counts, i.e., approximately 0.45 count per million reads (cpm) mapped, for at least 3 libraries across all 9 samples. Differences in library size were adjusted by the trimmed mean of M-values normalization method [64], implemented in the package edgeR [65]. Counts were transformed to log2 (cpm) values and associated precision weights and entered into the limma analysis pipeline [66]. Differentially expressed genes were identified by using generalized linear models that incorporate empirical Bayes methods to shrink the standard errors toward a common value, thereby improving testing power [67]. All sequencing data have been submitted to the Gene Expression Omnibus (GEO) and are available under accession number GSE203250.

## Confocal imaging

Cells were seeded and treated in μ-Slide 8 Well Glass plates (ibidi GmbH, Planegg, Germany) and visualized with Leica SP8-SMD confocal microscope (Leica Microsystems, Wetzlar, Germany) equipped with a 63 × 1.20 NA water-immersion objective lens. Images were acquired using 1,024 × 1,024 pixels with the pinhole set at 1 airy unit (AU). Excitation of the fluorescent probes was performed using white light laser (WLL, 50% laser output). Fluorescent emission was detected using an internal hybrid (HyD) detector. At least 3 entire images and 3 single-cell images were taken for each replicate. The experiments are a minimum of 2 biological duplicates.

Lipid droplet accumulation was measured on 3.7% formaldehyde fixed cells after 20-minute incubation with 2 μg/ml BODIPY 493/503 (Thermo Fisher Scientific, Massachusetts, USA) and mounted with Vectashield-H anti-fade medium (Vector Laboratories, California, USA). The WLL laser line (488 nm) was set at a laser power of 2.5%, and emission was detected selecting a spectral window of 505 to 550 nm.

ER was stained in live cells using ER Staining Kit—Red Fluorescence—Cytopainter (ab139482) (Abcam, Cambridge, UK) following the manufacturer's protocol. The principle of the stain reagent provided by the manufacturer is binding to the sulphonylurea receptors of ATP-sensitive K+ channels, which are prominent on the ER. Briefly, cells were stained with 1.5 μl/ml Detection Reagent and cultured with colorless DMEM for 45 minutes at 37˚C before detection by microscopy. The WLL laser line (596 nm) was set at a laser power of 5%, and emission was detected selecting a spectral window of 670 to 720 nm.

Lysosome were stained in live cells using LysoTracker Deep Red (Thermo Fisher Scientific, Massachusetts, USA) following the manufacturer's protocol. The LysoTracker probes consist of a fluorophore linked to a weak base that is only partially protonated at neutral pH. This allows LysoTracker probes to freely permeate cell membranes, enabling them to label live cells. LysoTracker probes are highly selective for acidic organelles. Briefly, cells were stained with 75 nM fluorescence probe and cultured with colorless DMEM for 60 minutes at 37˚C before detection by microscopy. The WLL laser line (647 nm) was set at a laser power of 5%, and emission was detected selecting a spectral window of 670 to 720 nm.

Early endosomes were stained by CellLight Early Endosomes-RFP, BacMam 2.0 Kit (Thermo Fisher Scientific, Massachusetts, USA). In brief, cells were incubated overnight with 40 μl/ml CellLight reagent, fixed with 3.7% formaldehyde, and mounted with Vectashield-H anti-fade medium. The WLL laser line (555 nm) was set at a laser power of 5%, and emission was detected selecting a spectral window of 565 to 700 nm. Human primary macrophages for co-staining assays were cultured overnight in fresh medium after the lipid treatment to allow for sufficient intracellular lipid transportation.

When performing co-staining of neutral lipids with ER, 2μg/ml BODIPY 493/503 and 1.5 μl/ml ER Detection Reagent were mixed in 1× Assay Buffy (equipped in ER kit). For lipid droplets-lysosome co-staining, the reagents were mixed in colorless DMEM. The co-staining duration followed the requirement of ER staining or lysosome staining.

All staining operations and confocal imaging were protected from light as much as possible.

## Flow cytometry analysis

Human primary macrophages (GM-CSF) were seeded in 24-well plates with a density as previously described and treated with 0.5 mM lipid emulsions for 6 hours. Approximately 1 μg/ml BODIPY 493/503 was used for cellular lipid droplet staining. After 20 minutes of incubation with BODIPY at 37˚C, cells were twice washed with PBS and then trypsinized. Samples were measured on a CytoFLEX cytometer (Beckman Coulter, Indianapolis, USA), and data were analyzed by FlowJo (BD, Oregon, USA).

## siRNA gene knock down assay

Silencing of *LPL*, *CAV1*, *CAV2*, *NPC1*, and *STARD3* in GM-CSF buffy coat human macrophages was carried out using ON-TARGETplus siRNA SMARTpool kits (Horizon Discovery Research company, Waterbeach, UK) following the instructions of the manufacturer. Briefly, GM-CSF macrophages were seeded in the desired plates at the density specified before and cultured overnight. Approximately 50 nM of siRNA was applied together with Lipofectamine

RNAiMAX Transfection Reagent (Thermo Fisher Scientific, Massachusetts, USA) for 48 hours. Real time-qPCR and immunoblotting were used to assess the transfection efficiency. The sequences of the siRNAs are listed in S1 Table.

## Statistical analysis

Data are presented as mean ± SD. Data analysis were performed using unpaired Student *t* test. GO cellular components analysis was completed by Enrichr [68–70]. All other plots on transcriptome data were generated by Graphpad Prism 8.

## Supporting information

**S1 Raw Data Checklist. Locations of raw data of figures in the paper.**
(XLSX)

**S1 Raw Images. Original blots for the western blotting figures in the paper.**
(PDF)

**S1 Raw Data. Original data of qPCR data, image J quantification, and other graphs.**
(ZIP)

**S1 Table. Target sequences for siRNA used in the study.**
(DOCX)

**S1 Fig. Silencing of LPL, CAV1, and CAV2 modulated the expression of cytokines altered by lipid treatment.** Human primary macrophages were treated with 1 mM VLDL-sized emulsion particles for 24 hours. The bar graphs were plotted as mean ± SD. Asterisk indicates significantly different in the marked comparisons according to Student *t* test. $^*p < 0.05$, $^{**}p < 0.01$. (The raw data can be found in "S1 Raw Data".)
(TIF)

**S2 Fig. Neutral lipids retained in lysosomes after inhibiting lysosomal acid lipase.** The figure illustrates single channel images of co-staining of lysosome (red) and neutral lipids (BODIPY 493/503, green) in human macrophages in the presence or absence of 30 μM Laslistat 2 ($n = 6$). Cells were treated with 0.5 mM VLDL-sized emulsion particles for 24 hours. Before imaging, cells were washed twice with PBS and cultured in fresh medium for 24 hours.
(TIF)

**S3 Fig. Neutral lipids retained in lysosomes of siNPC1-treated human primary macrophages.** The figure illustrates single channel images of co-staining of lysosome (red) and neutral lipids (BODIPY 493/503, green) in human macrophages treated with siCTRL or siNPC1 for 72 hours, followed by treatment with 0.5 mM VLDL-sized emulsion particles for 24 hours ($n = 6$). Before imaging, cells were washed twice with PBS and cultured in fresh medium for 24 hours.
(TIF)

**S4 Fig. Increased lipid accumulation in lysosomes by STARD3 silencing.** The figure presents single channel images for co-staining of lysosomes (red) and neutral lipids (BODIPY 493/503, green) in human macrophages treated with siCTRL or siSTARD3 for 48 hours and followed by treatment with 0.5 mM VLDL-sized emulsion particles for 24 hours ($n = 6$). Before imaging, cells were washed twice with PBS and cultured in fresh medium for 24 hours.
(TIF)

**S5 Fig. STARD3 silencing impairs accumulation of neutral lipids in the ER.** The figure presents single channel images for co-staining of ER (red) and neutral lipids (BODIPY 493/503, green) in human macrophages treated with siCTRL or siSTARD3 for 48 hours followed by treatment with 0.5 mM VLDL-sized emulsion particles for 24 hours ($n = 6$).
(TIF)

**S6 Fig. CHYL-sized emulsion particles promote lipid accumulation in cultured macrophages.** (a) The particle size distribution of CHYL-sized emulsion particles as determined by mastersizer 3000. (b) Mean fluorescence intensity (FITC-A) measured by flow cytometry of mouse RAW 264.7 macrophages treated with 1 mM CHYL-sized emulsion particles for 6 hours ($n = 3$). (c) mRNA expression of lipotoxic marker genes in RAW 264.7 macrophages. Bar graphs were plotted as mean ± SD. Statistical significance was analysed using 2-way ANOVA; $^*p < 0.05$, $^{**}p < 0.01$, $^{***}p < 0.001$, $^{***}p < 0.0001$. (The FACS data are available under repository ID FR-FCM-Z5K3. The raw data of bar graphs can be found in "S1 Raw Data".)
(TIF)

**S7 Fig. LPL mediates uptake of CHYL-sized emulsion particles in cultured macrophages.** (a) BODIPY 493/503 staining of intracellular neutral lipids in RAW 264.7 macrophages treated with 1 mM CHYL-sized emulsion particles for 6 hours in the presence or absence of 50 UI/ml human heparin ($n = 6$). (b) Quantification of the fluorescence images by ImageJ ($n = 4$). (c) BODIPY 493/503 staining of intracellular neutral lipids in human macrophages treated with siCTRL or siLPL for 48 hours followed by treatment with 0.5 mM CHYL-sized emulsion particles for 6 hours ($n = 6$). (d) Mean fluorescence intensity quantified by flow cytometry ($n = 3$). (e) mRNA expression of selected lipid-sensitive genes. (f) BODIPY 493/503 staining of intracellular neutral lipids in human macrophages treated with siCTRL or siLPL for 48 hours followed by treatment with 0.5 mM human plasma isolated CHYL for 6 hours ($n = 6$). (g) Mean fluorescence intensity quantified by Image J ($n \geq 4$). The bar graphs were plotted as mean ± SD. Asterisk indicates significantly different from control according to Student $t$ test. $^*p < 0.05$, $^{**}p < 0.01$, $^{***}p < 0.001$. (The FACS data are available under repository ID FR-FCM-Z5KY. The raw data of bar graphs can be found in "S1 Raw Data".)
(TIF)

**S8 Fig. The C-terminal portion of LPL mediates uptake of CHYL-sized emulsion particles in cultured macrophages.** (a) BODIPY 493/503 staining of RAW 264.7 macrophages treated with 1 mM CHYL-sized emulsion particles for 6 hours in the presence or absence of 0.2 μM of the catalytic LPL inhibitor GSK264220 ($n = 6$). (b) Mean fluorescence intensity quantified by Image J ($n \geq 4$). (c) BODIPY 493/503 staining of human primary macrophages treated with 0.5 mM CHYL-sized emulsion particles for 6 hours in the presence or absence of antibody F1 targeting the N-terminal portion of LPL (2 μg/ml) ($n = 6$). (d) Mean fluorescence intensity quantified by Image J ($n \geq 4$). (e) BODIPY 493/503 staining of human primary macrophages treated with 0.5 mM CHYL-sized emulsion particles for 6 hours in the presence or absence of antibody 5D2 targeting the C-terminal portion of LPL (2 μg/ml) ($n = 6$). (f) Mean fluorescence intensity quantified by Image J ($n \geq 4$). The bar graphs were plotted as mean ± SD. Asterisk indicates significantly different from control according to Student $t$ test. $^{****}p < 0.0001$. (The raw data of bar graphs can be found in "S1 Raw Data".)
(TIF)

**S9 Fig. CHYL-sized emulsion particles are taken up by macrophages via caveola-mediated endocytosis.** (a) Early endosome staining of RAW 264.7 macrophages treated with 1 mM of CHYL-sized emulsion particles for 6 hours ($n = 6$). (b) BODIPY 493/503 staining of RAW

264.7 macrophages treated with 1 mM CHYL-sized emulsion particles for 6 hours in the presence or absence of 10 μg/ml chlorpromazine or 200 μM genistein ($n = 6$). (c) Mean fluorescence intensity quantified by Image J ($n = 6$). (d) BODIPY 493/503 staining of human primary macrophages treated with 0.5 mM CHYL-sized emulsion particles for 6 hours in the presence or absence of 200 μM genistein ($n = 6$). (e) Mean fluorescence intensity quantified by Image J ($n = 3$). The bar graphs were plotted as mean ± SD. ***$p < 0.001$. (The raw data of bar graphs can be found in "S1 Raw Data".)
(TIF)

**S10 Fig. Silencing of Caveolin 1 and 2 impairs uptake of CHYL-sized emulsion particles by macrophages.** (a) BODIPY 493/503 staining of human macrophages treated with siCTRL, siCAV1, or siCAV2 for 48 hours followed by treatment with 0.5 mM CHYL-sized emulsion particles for 6 hours ($n = 6$). (b) Mean fluorescence intensity quantified by flow cytometry ($n = 3$). (c) BODIPY 493/503 staining of human macrophages treated with siCTRL, siCAV1, or siCAV2 for 48 hours followed by treatment with 0.5 mM human plasma-isolated CHYL for 6 hours ($n = 6$). (d) Mean fluorescence intensity quantified by Image J ($n = 4$). The bar graphs were plotted as mean ± SD. Asterisk indicates significantly different from control according to Student $t$ test. *$p < 0.05$. (The FACS data are available under repository FR-FCM-Z5JV and FR-FCM-Z5KZ. The raw data of bar graphs can be found in "S1 Raw Data".)
(TIF)

**S11 Fig. TG in CHYL-sized emulsion particles are degraded by lysosomal acid lipase.** (a) BODIPY 493/503 staining of human macrophages treated with 0.5 mM CHYL-sized emulsion particles for 6 hours in the presence or absence of 30 μM Lalistat 2 ($n = 6$). (b) Mean fluorescence intensity quantified by Image J ($n = 6$). (c) Heatmaps showing changes in the expression of genes involved in lysosome activity in RAW 264.7 macrophages treated with 0.5 mM CHYL-sized emulsion particles for 6 hours ($p < 0.01$, SLR > 1). Scale bar depicts SLR. (The raw data of RNA-sequencing are available under accession number GSE203250. Other data can be found in "S1 Raw Data".)
(TIF)

## Acknowledgments

The authors would like to thank Chang Sun for helping with the screening of chemical inhibitors of endocytosis, Montserrat de la Rosa Rodriguez and Qi Zheng for helping with confocal microscope technology, and Benthe van der Lugt for helping with human monocytes isolation. We thank Anne Beigneux (UCLA) for providing the LPL antibody 5D2.

## Author Contributions

**Conceptualization:** Lei Deng, Frank Vrieling, Rinke Stienstra, Anouk L. Feitsma, Sander Kersten.

**Data curation:** Lei Deng, Guido J. Hooiveld, Sander Kersten.

**Formal analysis:** Lei Deng, Frank Vrieling, Guido J. Hooiveld.

**Funding acquisition:** Sander Kersten.

**Investigation:** Lei Deng, Anouk L. Feitsma.

**Methodology:** Lei Deng, Frank Vrieling.

**Project administration:** Sander Kersten.

**Supervision:** Rinke Stienstra, Anouk L. Feitsma, Sander Kersten.

**Visualization:** Lei Deng.

**Writing – original draft:** Lei Deng.

**Writing – review & editing:** Frank Vrieling, Rinke Stienstra, Guido J. Hooiveld, Anouk L. Feitsma, Sander Kersten.

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
