## [Editor Report · Decision Letter 0]

15 Dec 2021

Dear Dr Kersten, 

Thank you for submitting your manuscript entitled "Caveolae mediated endocytosis of VLDL particles in macrophages requires NPC1 and STARD3 for further lysosomal processing" for consideration as a Research Article by PLOS Biology.

Your manuscript has now been evaluated by the PLOS Biology editorial staff as well as by an academic editor with relevant expertise and I am writing to let you know that we would like to send your submission out for external peer review.

Once your full submission is complete, your paper will undergo a series of checks in preparation for peer review. Once your manuscript has passed the checks it will be sent out for review. To provide the metadata for your submission, please Login to Editorial Manager (https://www.editorialmanager.com/pbiology) within two working days, i.e. by Dec 17 2021 11:59PM.

If your manuscript has been previously reviewed at another journal, PLOS Biology is willing to work with those reviews in order to avoid re-starting the process. Submission of the previous reviews is entirely optional and our ability to use them effectively will depend on the willingness of the previous journal to confirm the content of the reports and share the reviewer identities. Please note that we reserve the right to invite additional reviewers if we consider that additional/independent reviewers are needed, although we aim to avoid this as far as possible. In our experience, working with previous reviews does save time. 

If you would like to send previous reviewer reports to us, please email me at ialvarez-garcia@plos.org to let me know, including the name of the previous journal and the manuscript ID the study was given, as well as attaching a point-by-point response to reviewers that details how you have or plan to address the reviewers' concerns. 

Given the disruptions resulting from the ongoing COVID-19 pandemic, please expect some delays in the editorial process. We apologise in advance for any inconvenience caused and will do our best to minimize impact as far as possible.

Kind regards,

Ines

--

Ines Alvarez-Garcia, PhD

Senior Editor

PLOS Biology

---

## [Decision Letter · Decision Letter 1]

16 Feb 2022

Dear Dr Kersten,

Thank you for submitting your manuscript entitled "Caveolae mediated endocytosis of VLDL particles in macrophages requires NPC1 and STARD3 for further lysosomal processing" for consideration as a Research Article at PLOS Biology. Thank you also for your patience as we completed our editorial process, and please accept my apologies for the delay in providing you with our decision. Your manuscript has been evaluated by the PLOS Biology editors, an Academic Editor with relevant expertise, and by two independent reviewers.

You will see that the reviewers find the conclusions of the manuscript interesting and novel, but they also raise several issues that need to be addressed. Reviewer 1 has some concerns regarding the difference in composition of the VLDL-mimic particles used in the study and would like to see additional evidence to confirm the source of the FFAs in media. Reviewer 2 has similar concerns and also highlights some concerns regarding the use of the 5D2 antibody that should be addressed.

In light of the reviews (attached below), we will not be able to accept the current version of the manuscript, but we would welcome re-submission of a revised version that takes into account the reviewers' comments. We cannot make any decision about publication until we have seen the revised manuscript and your response to the reviewers' comments. Your revised manuscript is also likely to be sent for further evaluation by the reviewers.

We expect to receive your revised manuscript within 3 months. 

**IMPORTANT - SUBMITTING YOUR REVISION**

3. Resubmission Checklist

a) *PLOS Data Policy*

b) *Published Peer Review*

Sincerely,

Ines

--

Ines Alvarez-Garcia, PhD

Senior Editor

PLOS Biology

Reviewers' comments

Rev. 1: Alison Kohan – note that this reviewer has signed her review

Overall, the hypotheses tested in this manuscript are impactful and suggest serious reconsideration of the triglyceride metabolic pathways in macrophages and how these may be functioning in diseases like atherosclerosis. The idea that macrophages have a dedicated FFA secretion process that is part of their normal uptake/metabolism of triglyceride-rich lipoproteins is intriguing. The authors use VLDL-sized casein-emulsified particles (which they call "VLDL-mimic" particles) to study the uptake and intracellular metabolism of neutral lipids in both RAW and human macrophages. In a few instances, the authors also confirm their finding using human VLDL. The authors use chemical and siRNA inhibition to block key steps in endocytosis (LPL activity and caveolin), transport to and hydrolysis in the lysosome (LAL, NPC1), and movement between organelles (using stains for lysosome, nucleus, neutral lipid) and secretion into the media (NEFA assays). Overall, the authors conclude that emulsion particle-derived lipids can be endocytosed, transported to the lysosome, and when normal metabolism to/from the lysosome is blocked (via NPC1 and STARD3), the FFA can't be secreted nor travel to the ER. The authors have developed an interesting hypothesis and I agree with their assessment that the mechanisms governing metabolism of lipoprotein TAG by immune cells is underdeveloped. However, I have some criticisms of the way the quite of experiments were carried out.

(1) Other than sharing a similar size, the VLDL-sized emulsion particles (coated with casein) do not share significant similarities to physiological VLDL. The emulsion particles have no cholesterol, no apolipoproteins, and the phospholipid source is unique from VLDL. These particles are more like milk than VLDL. The Discussion section acknowledges the drawbacks of using VLDL-mimic particles, and the authors have fully and clearly described these drawbacks. They also used human VLDL to duplicate their finding that LPL is required. This lends some confidence. However, the authors haven't duplicated other key findings using this approach. Considering this issue, I would caution against calling them VLDL-mimics. My opinion would be to remove the term "VLDL" or "VLDL-mimic", and instead adjust the title, figures, and annotations to refer to "emulsion particles" or rather than VLDL.

(2) Some details about methods and experiments appear to be missing. In the Methods section there is not enough information about how the emulsion particles were made, and why casein was chosen as the emulsifier. How was the lipid dried, solubilized, vortexed, etc.? There could also be more information about the co-stains used for neutral lipids versus lysosomes. Is there evidence for the specificity of the lysosomal stains (versus cross-staining of ER)? Finally, I can't find the experimental number (n=? in each figure legend). Assuming the n is reasonable, the statistics seem reasonably done and interpreted. Figure 8d is referred to in the text but isn't described in its figure legend. Critically, this figure purports to show transit from the lysosome to ER, but it's unclear whether the stain is still the lysosomal stain or an ER-specific stain.

(3) In some cases, the data could be interpreted in a variety of ways. For example, when NPC1 is inhibited and NEFA rises in the media, this doesn't necessarily mean that the FFA is being actively secreted from intracellular pools within the macrophage. It could mean that there is a change in the rate of FFA generation in the media (from VLDL precursor), or that FFAs (derived from VLDL in the media) are being retained in the media rather than being endocytosed. It would be helpful for this to be discussed. Similarly, because the authors use a neutral lipid stain it is hard to dissect differences between the accumulation of triglycerides and cholesterol, and in some cases the authors conclude that the stains show the movement of FFA between cellular locations. Without specific radio- or fluorescent labels, or more information about the variety of stains used, I'm unsure about how to interpret this. They do bring this issue up in the Discussion, but this concern should probably be a part of the Results as well.

In summary, despite these concerns, the findings in this manuscript add to our knowledge about extracellular triglyceride in regulating intracellular metabolism in macrophages. After reading the manuscript I kept thinking about how it might impact my own and others' work - so there is certainly potential impact. In my opinion, this manuscript (with some revisions) should be published.

Rev. 2:

I have read with interest the manuscript of Lei Deng, et al.

The manuscript shows that uptake of VLDL by the macrophage is mediated by the C terminal of LPL via caveolar-mediated endocytosis. Interestingly, NPC1 was found to promote the extracellular efflux of fatty acids from lysosomes, while Stard3 is involved in the transfer of lysosomal fatty acids to the ER for subsequent storage as triglycerides. I feel that authors' manuscript is well performed and contains interesting data from a basic research perspective. However, in my point of view, several limitations will negatively affect the study and some claims are overstated based on their evidence.

The main issue about this manuscript:

1. Much of the cited literature that is used to make the case for their hypothesis is very old and actually refers to macrophage uptake the VLDL and LPL serve as the bridge for that.

2. To my recollections, several papers mentioned that 5D2 binds to human LPL but fails to bind to mouse LPL. Because RAW 264.7 cells were derived from BALB/c mice and only express mouse LPL. Therefore, it will be confused to interpret the data (Figure 3e) that 5D2, definitely failed to block the C-terminal of mouse LPL, but impaired the vLPLm uptake on mouse macrophages. It cannot be the strong support for the main idea. Thus, the statement that C-terminal region of mouse LPL is required for macrophage uptake of vLDLm-TG is not supported by the data

3. Detecting the medium FFA and genes expression levels cannot be the strong support for that NPC1 was found to promote the extracellular efflux of fatty acids from lysosomes, while Stard3 is involved in the transfer of lysosomal fatty acids to the ER for subsequent storage as triglycerides.

4. The authors treated the RAW264.7 macrophages with 50 UI/ml of heparin for 2 hours, LPL protein levels decreased by ~20%. Several groups demonstrated that LPL can be internalized into cells, it is highly recommended to perform the immunofluorescent staining or live cell fluorescence binding experiment rather than Western blot, if you want to test the release effect of heparin on LPL.

5. The authors explained that "the LPL remaining after heparin treatment likely represents the functionally inactive intracellular LPL pool". Authors should present their data to support this idea. Previous studies showed that high levels of heparin (500U/ml) fully dissociate LPL from cell surface. We suspected that the authors could only release some proportion of LPL, the remaining LPL might be still functional.

6. In Figure2D, the LPL bands were smeared. This data is even less convincing.

7. Silencing of both Caveolin 1 and 2 will impair uptake of VLDL further, compared with single siRNA treatment?

8. Did BODIPY signals co-localize with cav-1 or cav-2 on macrophages loaded with vLDLm?

9. The author should also present single-channel images of BODIPY and lysosome marker staining on Figure 6D and Figure 7E. Otherwise, it will be difficult to interpret those data. In Figure 6d left panel, could authors observe the colocalization of BODIPY and lysosome marker?

10. In Figure 7E, right panel, it showed heterogenetic co-localization on macrophages treated with siNPC1 followed by VLDL treatment? Please explain that. We recommended to knockout those genes via Crispr. RNA interference and inhibitors did not become clear to me since low transfection efficacy and low specificity might negatively affect the results.

11. Please add the scale bar to the immunofluorescence images clearly.

12. In supplementary Figure S1G, the authors demonstrated that vLPLm or CHYLm treatment on macrophages induced the expression levels of ER stress inflammatory response related genes, did you observe the similar trend of cytokine expression (i.e., TNF-a, IL-1b, MCP-1, MCP-1, and MMP-3)? Did LPL siRNA, genistein, siCAV1, siCAV2 or 5D2 treatment rescue the phenotype?

13. The figure citation in the manuscript is in ascending numerical order. The authors should citate the figure S1a first.

14. Ref 13 and ref 19 are duplicated.

15. siRNA sequences targeted for LPL, CAV1, CAV2, NPC1 and STARD3 should be required.

---

## [Decision Letter · Decision Letter 2]

21 Jun 2022

Dear Dr Kersten,

Thank you for your patience while we considered your revised manuscript entitled "Caveolae-mediated endocytosis of VLDL-sized emulsion particles in macrophages requires NPC1 and STARD3 for further lysosomal processing" for publication as a Research Article at PLOS Biology. This revised version of your manuscript has been evaluated by the PLOS Biology editors, the Academic Editor and two of the original reviewers.

Based on the reviews (attached below), we are likely to accept this manuscript for publication, provided you consider the remaining points raised by Reviewer 2. Please also make sure to address the following data and other policy-related requests stated below.

In addition, we would like you to consider a suggestion to improve the title:

"Macrophages take up VLDL-sized emulsion particles through caveolae-mediated endocytosis and excrete part of the internalized triglycerides as fatty acids"

We expect to receive your revised manuscript within two weeks. 

*Published Peer Review History*

*Press*

Sincerely,

Ines

--

Ines Alvarez-Garcia, PhD

Senior Editor

PLOS Biology

DATA POLICY:

Thank you for providing a file including all raw data underlying the graphs shown in the figures. Please indicate what dataset belongs to each of the figures in the excel file and also where the data can be found in the figure legends. We need you to provide the data underlying the following figures:

Fig. 1A, C-I; Fig. 2C, F, G, I; Fig. 3B, D, F; Fig. 4C, E; Fig. 5A, B, D, F; Fig. 6A, C, E, F; Fig. 7A, D, F-H; Fig. 8A, B, E; Fig. S1A, B; Fig. S2A, B; Fig. S4A, B; Fig. S5A-F; Fig. S6A, B; Fig. S7A-D; Fig. S8 and Fig. S10

In addition, for figures containing FACS data, we ask that you provide FCS files and a picture showing the successive plots and gates that were applied to the FCS files to generate the figure. We would suggest you deposit the data in the flow repository (http://flowrepository.org/).

Many thanks for providing the original, uncropped raw images of all blot and gel results reported. Please indicate in the file to which figures correspond each of the images. You can read our guidelines for how to prepare and upload this data: https://journals.plos.org/plosbiology/s/figures#loc-blot-and-gel-reporting-requirements

Reviewers' comments:

Rev. 1: Alison Kohan

I am totally satisfied with the revised manuscript. I think the work is very well done, clearly communicated, and the main conclusion rigorously determined. 

Rev. 2: 

For the response to the question 1 from review 1, I prefer to refer to "emulsion particles" as reviewer1 suggested rather than "VLDLsep" which may cause readers' misunderstanding. As suggested by reviewer1, the emulsion particles have no cholesterol, no apolipoproteins, and the phospholipid source is unique from VLDL.

For the response to the question 2 from review1, it seems that the author did not response to "It could mean that there is a change in the rate of FFA generation in the media (from VLDL precursor), ".

I stay neutral for this article.

---

## [Editor Report · Decision Letter 3]

9 Jul 2022

Dear Dr Kersten,

Thank you for the submission of your revised Research Article entitled "Macrophages take up VLDL-sized emulsion particles through caveolae-mediated endocytosis and excrete part of the internalized triglycerides as fatty acids" for publication in PLOS Biology. On behalf of my colleagues and the Academic Editor, Rebecca Haeusler, I am happy to say that we can in principle accept your manuscript for publication, provided you address any remaining formatting and reporting issues. These will be detailed in an email you should receive within 2-3 business days from our colleagues in the journal operations team; no action is required from you until then. Please note that we will not be able to formally accept your manuscript and schedule it for publication until you have completed any requested changes.

PRESS

Sincerely, 

Ines

--

Ines Alvarez-Garcia, PhD

Senior Editor

PLOS Biology
